

# 1 The Probability Distribution of Daily Precipitation at the Point and
# 2 Catchment Scales in the United States

Lei Ye[1*], Lars S. Hanson[2], Pengqi Ding[1], Dingbao Wang[3], Richard M.Vogel[4]
1 School of Hydraulic Engineering, Dalian University of Technology, Dalian, China
2 Institute for Public Research, Center for Naval Analyses, Arlington, Virginia, USA.
3 Department of Civil and Environmental Engineering, Tufts University, Medford, Massachusetts, USA
4 Department of Civil, Environmental, and Construction Engineering, University of Central Florida,
Orlando, Florida, USA
**Abstract:** Choosing a probability distribution to represent daily precipitation depths is
important for precipitation frequency analysis, stochastic precipitation modeling and in
climate trend assessments. Early studies identified the 2-parameter Gamma (G2)
distribution as a suitable distribution for wet-day precipitation based on traditional
goodness of fit tests. Here, probability plot correlation coefficients and L-moment
diagrams are used to examine distributional alternatives for the full-record and wet-day
series of daily precipitation at the point and catchment scales in the United States.
Importantly, the G2 distribution performs poorly in comparison to either the Pearson
Type-III (P3) or Kappa (KAP) distributions. The analysis indicates that the P3
distribution fits the full record of daily precipitation at both the point and catchment
scales remarkably well; while the KAP distribution best describes the distribution of wet-
day precipitation at the point scale, and the performance of KAP and P3 distributions is
comparable for wet-day precipitation at the catchment scale.
**Key Words:** Climate; Rainfall; Weather; L-moment diagram; PPCC; Pearson type III;
Kappa; Gamma; Wet-day; Frequency analysis; Trend detection; Stochastic weather
models

# 30 1. Introduction

Establishing a probability distribution that provides a good fit to daily
precipitation depths has long been a topic interest in the fields of hydrology, meteorology,
and others. The investigations into the daily precipitation distribution are primarily
spread over three main research areas, namely, (1) stochastic precipitation models, (2)
frequency analysis of precipitation, and (3) precipitation trends related to global climate
change. Table 1 displays a sampling of the literature in these three fields, the particular
precipitation series and durations under investigation, and the proposed distributions
identified. Table 1 is by no means exhaustive; it only attempts to document the
widespread interest in the determination of a suitable distribution for daily precipitation
totals for various purposes.

[*Table* 1 *goes here*]

# 42 1.1 Stochastic precipitation and climate models:

* Corresponding author. E-mail address: yelei@dlut.edu.cn



The first section in Table 1 presents a small portion of the literature related to
stochastic precipitation modeling also referred to as stochastic weather modeling.  The
purpose of such models is not so much to investigate the properties of precipitation, but
instead to produce artificially generated precipitation sequences that can be used as inputs
to other models to explore the behavior of hydrologic systems (Buishand, 1978;Waymire
and Gupta, 1981).  A wide range of types of stochastic precipitation generators exist as
evidenced from review articles Waymire and Gupta (1981), Wilks and Wilby (1999),
Srikanthan and McMahon (2001) and Chen and Brissette (2014).  Also see the
introduction of Mehrotra et al. (2006) for a nice review.

Since our central goal is to select a suitable generalized probability distribution
for modeling daily precipitation depths, we are only concerned with the class of "two-
part" stochastic daily precipitation models that utilize a probability distribution function
to describe precipitation amounts on wet-days, while precipitation occurrence is
separately described using a Markov model or some form of a stochastic renewal process
(Buishand, 1978;Geng et al., 1986;Waymire and Gupta, 1981;Watterson, 2005).

It is evident from Table 1 that the wet-day precipitation series is virtually the only
daily precipitation series that is even considered in the stochastic precipitation model
literature.  Thom's (1951) suggestion of the 2-parameter Gamma (G2) distribution
function for wet-day amounts seems to carry considerable weight.  Following the
suggestion of numerous previous authors, both Watterson and Dix (2003) and Watterson
(2005) assumed a Gamma distribution for wet-day rainfall in the development of
stochastic rainfall models.. Buishand (1978) lent support to the suggestion of the Gamma
distribution by showing that for the wet-day series at six stations, the empirical
Coefficient of Variation to Coefficient of Skewness ratio was quite close to the
theoretical value of two for a Gamma distribution.

Geng et al. (1986) used a simple regression to show that the beta parameter of the
Gamma distribution for a given month can be predicted reasonably well by the average
rainfall per wet-day in that month.  Geng et al. (1986) also provided a good review of
other literature supporting the use of the Gamma distribution for modeling wet-day
rainfall.

While the G2 distribution is by far the most preferred distribution for wet-day
precipitation amounts, other distributions have also been suggested.  Woolhiser and
Roldan (1982) and Wilks (1998) both suggested the use of a three-parameter mixed
exponential distribution instead of G2.  The three-parameter exponential distribution can
describe wet-day amounts by mixing two distinct exponential distributions (each with its
own mean parameter) with a parameter that chooses which one to use.  Through a variety
of goodness of fit tests and log-likelihood analyses, the mixed exponential is shown as
being preferred to G2 (Wilks, 1998).

The Weibull (W2) and to a lesser extent the exponential distribution have also
been suggested for modeling daily precipitation amounts (Duan et al., 1995;Burgueno et
al., 2005).  Duan et al. (1995) used a Chi-squared test to demonstrate that synthetic
rainfall generated from the Weibull and Gamma (with parameters estimated by method of
moments) models best matches the observed data within each month.  Separate models



were created for each calendar month.  (Burgueno et al., 2005) used graphical methods
and the Kolmogorov-Smirnov test to give support to the W2 and exponential distributions.

## 1.2 Precipitation frequency analysis:

The second section of Table 1 displays a small portion of the literature related to
precipitation frequency analyses.  Extreme values of rainfall are of particular interest to
urban planners, engineers and hydrologists working on problems related to storm
drainage, flooding, and other natural hazards such as precipitation-induced slope failures
(landslides).  Precipitation frequency analyses are one way to generate the necessary
precipitation totals at given return period for hydraulic design purposes.  A key step in
frequency analysis of precipitation involves selection of a suitable distribution for
representing precipitation depths to investigate the extremes.  While these analyses can
be conducted for multiple precipitation durations, we focus on those that investigate the
1-day duration.
As the extreme rainfall values are of primary importance in these studies, a highly
censored series of rainfall is often useful in these analyses.  The Annual Maximum Series
(AMS) and Partial Duration Series (PDS) are often used in hydrologic frequency
investigations (Stedinger et al., 1993).  Table 1 displays that many of the precipitation
frequency investigations of daily precipitation depths have selected the AMS series.  The
wet-day series is actually a PDS with zeros and values lower than the detection limit of
the instrument (i.e., "trace" values) censored.
In perhaps the most comprehensive assessment of the distribution of precipitation
extremes, Papalexiou and Koutsoyiannis (2013) examined the goodness-of-fit of the
GEV distribution to a global dataset of AMS at 15,137 sites with lengths varying from 40
to 163 years.  Analysis of such a large dataset enabled them to conclude that GEV models
of AMS series of daily precipitation provide a good approximation with the shape
parameter depending critically upon both the location and length of the series under
consideration. Interestingly, when record length and location are taken into account, the
shape parameter appears to exhibit a relatively narrow range of small positive values.
For many years, the most common approach to summarizing precipitation
frequency analyses in the United States was the work of Hershfield (1961), which is
commonly referred to as TP-40.  Hershfield (1961) fitted a Gumbel distribution to the
AMS series of 24-hour precipitation.  More recently investigators have completed these
types of analyses by using the method of L-moments and other methods that are more
powerful than the traditional goodness of fit measures.  In the context of a national
revision to the TP-40 rainfall frequency atlas and after the application of L-moment
goodness-of-fit evaluations, Bonnin et al., (2006) fitted a generalized extreme value
(GEV) distribution to the AMS of rainfall.
Bonnin et al. (2006) performed a very comprehensive national assessment of
precipitation frequency by applying the most up-to-date developments in regional
frequency analysis to series of annual maximum n-minute precipitation.  Using both at-
site and regional L-moment goodness-of-fit results, climatic considerations and
sensitivity testing, the GEV distribution was selected to best represent the underlying
distributions of all daily and hourly AMS rainfall data. GEV was also selected for the 5-,
10-, and 15-minute AMS rainfall data.  Naghavi and Yu (1995) also chose the GEV for a



study of rainfall extremes in Louisiana.  Similarly, Lee and Maeng (2003) selected the
GEV and the generalized logistic distributions based on L-moment analysis of 58 stations
in Korea.
While the results of Bonnin et al. (2006) apply to the United States, other authors
have found similar results using similar methods in other parts of the world.  Pilon et al.
(1991) used L-moment goodness-of-fit results to show that the Gumbel distribution
should be rejected in the favor of the GEV in Ontario, Canada.  In Korea, Park and Jung
(2002) successfully used the Kappa distribution (of which the GEV is a special case) to
generate extreme precipitation quantile maps using both Maximum Likelihood
Estimators (MLE) and L-moment estimators (L-ME) for Kappa parameter estimation.
They found convergence failure at some stations for the L-ME, and lack of fit for those
series fit with MLEs when sample size was too small.
Interestingly, while a great deal of attention is given to fitting distributions to the
relatively short AMS series of precipitation depths, very few studies directly explore the
probability distribution of the complete series of daily precipitation (including zeros) or
the wet-day series of daily precipitation (zeros excluded).  Shoji and Kitaura (2006)
investigated both full-record and wet-day daily precipitation series, but included only the
normal, lognormal, exponential, and Weibull distributions as candidate distributions, and
did not employ modern regional hydrologic methods such as the method of L-moments.
Perhaps the most thorough investigations, to date, on the probability distribution
of daily precipitation amounts are the global studies by Papalexiou and Koutsoyiannis
(2012, 2016).  Papalexiou and Koutsoyiannis (2012) derived a generalized Gamma
distribution (GG) from Entropy theory, using plausible constraints for wet-day series of
daily precipitation series. Together, the two studies by Papalexiou and Koutsoyiannis
(2012, 2016) revealed that the GG distribution provides a good approximation to the
behavior of observed L-moments of global series of wet-day daily precipitation at 11,519
and 14,157 stations, respectively.
Deidda and Puliga (2006) investigated the degree of left-censoring of wet-day
series needed to fit a Generalized Pareto (GPA) distribution for 200 stations in Sardinia,
Italy with a range of modern statistical analysis techniques.  The "failure-to-reject"
goodness-of-fit method was used to establish an optimal threshold for left censoring at
each station to make the observed data fit a GPA distribution.  Often, Deidda and Puliga
(2006) found that no optimal threshold for left censoring could make the data fit a GPA
distribution at 5 and 10% confidence intervals.  Deidda and Puliga (2006) remarked that
data rounding off may explain some of the lack of fit, but their results still leave room for
debate on the most likely candidate distribution for daily precipitation.

### 1.3 Precipitation trends and changes:

The third section of Table 1 summarizes a small portion of the precipitation trend
literature which has become a rather large area of inquiry due to concerns over climate
change, as evidenced from recent reviews on the subject (Easterling et al.,
2000;Trenberth, 2011;Madsen et al., 2014).  Interestingly, within the literature devoted to
detection of changes in precipitation patterns, we find a reliance on previous studies of
the probability distribution of daily precipitation for evaluating changes in distributional
parameters and in selecting candidate distributions.  Almost universally, the G2





distribution appears to be accepted without serious consideration of alternative
distributions. For instance, (Groisman et al., 1999) wrote simply, "It is widely
recognized that the distribution of daily precipitation totals, P, can be approximated by
the Gamma distribution." That is not to say the G2 distribution is not tested for its fit to
the observed data. For instance, (Groisman et al., 1999) compared maps of the empirical
probability of summer 1-day rainfall exceeding 50.4 mm with maps of probabilities
determined by a stochastic model using the fitted G2 distribution for the amounts. They
found acceptable fits in regions where there are enough observed daily rainfall events
greater than 50.4 mm.
This is an interesting contrast to the precipitation frequency analysis literature
where a Gamma distribution is often fit to wet-day series for the purpose of examining
extreme rainfalls instead of using the AMS series fitted by a GEV or other distribution.
Yoo et al. (2005) explained that conventional frequency analysis (using AMS) cannot
expect to predict precipitation changes resulting from climate change; while an
examination of the differences in the Gamma distribution's parameters (fitted to the
whole wet-day record) might predict such changes. They found that modifying the
parameters of the daily Gamma distribution can explain changes in rainfall quantiles
predicted by General Circulation Models (GCM) under various climate change scenarios.
Wilby and Wigley (2002) plotted the expected 100-year changes in the shape and scale
parameter of the G2 distribution according to two GCM models' predictions.
In a national study of precipitation trends, Karl and Knight (1998) employed the
G2 distribution to fill in missing precipitation observations. Karl and Knight (1998) wrote
that "To determine if precipitation occurs on any missing day, a random number
generator is used such that the probability of precipitation is set equal to the empirical
probability of precipitation during that month. If precipitation occurs, then the gamma
distribution is used to determine the amount that falls for that day, again using a random
number generator." Both Watterson and Dix (2003) and Watterson (2005) assumed a G2
distribution for daily precipitation in the development of stochastic rainfall models for
use in evaluating changes in precipitation extremes.
We conclude from this brief review that both the precipitation trend and climate
change literature have widely used the G2 distribution as a powerful tool to examine not
only the possible changes in precipitation patterns, but also the relative rate of change in a
geospatial context through mapping. In summary, there are a wide variety of previous
studies which have explored the probability distribution of daily precipitation for the
purposes of precipitation frequency analysis, stochastic precipitation modeling and for
trend detection. There seems to be a consensus that annual maxima appear to be well
approximated by either a GEV, Gumbel or Gamma probability density function (pdf) and
that series of wet-day daily precipitation totals are well approximated by a Gamma
Generalized Gamma, or in some cases a mixed exponential pdf. However, other than the
two recent global studies by Papalexiou and Koutsoyiannis (2012, 2016). We are
unaware of any studies that have used recent developments in regional hydrologic
frequency analysis such as L-moment diagrams or probability plot goodness of fit
evaluations to evaluate the probability distribution of the *complete series* of daily
precipitation.





The recent studies by Papalexiou and Koutsoyiannis (2012, 2016) represent
perhaps the most comprehensive studies to date, however, they only consider wet-day
series of daily precipitation and their L-moment evaluations only evaluate the
relationship between L-Skewness and L-Cv, thus they were unable to fully evaluate the
goodness-of-fit of the several relatively new three-parameter pdfs introduced in their
studies such as the generalized Gamma (GG) and the generalized Burr type XII (GB)
pdfs which would require construction of L-Kurtosis versus L-Skew diagrams.
Analogous to those two studies, this paper uses several large scale national datasets to re-
examine the question of which of the commonly used continuous distribution functions
which are widely used in the fields of hydrology, meteorology and climate best fit both
wet-day and complete series of observed daily precipitation data.
Instead of considering the GG distribution, the pdf recommended by both
Papalexiou and Koutsoyiannis (2012, 2016), which is only suited to wet-day series, has
seen very limited use and for which analytical and/or polynomial relationships for L-
Kurtosis are unavailable as they are for most commonly used pdfs in hydrology, we
consider the more widely used 3 parameter generalization of the Gamma distribution
known as the Pearson type III (P3) distribution. Once analytical and polynomial L-
moment relationships and parameter estimation methods become available for the GG
distribution, future studies should compare the P3 and GG distributions on wet-day series,
because on the basis of this study, and Papalexiou and Koutsoyiannis (2016), the P3 and
GG distributions appear to have tremendous potential for approximating the distribution
of wet-day series.
Our primary objective is to use a very large spatially distributed dataset at both
the point and catchment scales, to determine a suitable probability distribution of full-
record series and wet-day series of daily precipitation using L-moment diagrams and
probability plot correlation coefficient goodness of fit statistics. Analogous to the recent
study by Papalexiou and Koutsoyiannis (2016), these evaluations yield very different
conclusions than previous research on this subject.

## 2. Study area and data

Precipitation depths at the point and catchment scales are important information
in hydrology, meteorology, and other fields, thus our study focuses on both of them. For
point precipitation, we employ a data set comprised of daily precipitation depths at 237
first-order NOAA stations from 49 U.S. states (Hawaii is excluded due to fundamentally
different precipitation behavior). Station locations are shown in Figure 1a. In contrast,
the areal average precipitation for 305 catchments in the international Model Parameter
Estimation Experiment (MOPEX) data set (Duan et al., 2006) is also selected for analysis.
The catchment locations and boundaries are shown in Figure 1b. The data were quality
controlled to remove null values. When greater than 6 null values occurred in a given
year or greater than 3 in a given month, the full year of data was removed. When fewer
than these numbers of null values were present, they were treated as zeroes. The average
record length for point precipitation depths for the 237 sites is 24,657 days (67.5 years).
The distribution of record lengths corresponding to the 237 first-order NOAA stations is
shown in Figure 2. The MOPEX data set consists of 56 years of areal average





precipitation from 1948 to 2003, corresponding to a fixed record length 20,454 days for
each of the 305 catchments shown in Figure 1b.
[*Figure* 1 *goes here*]
[*Figure* 2 *goes here*]
In addition to the full-record series of daily precipitation, wet-day series were
extracted from both data sets. The wet-day series were constructed by excluding zero
and "trace" values (those with less than "0.01" recordable precipitation). Wilks (1990)
discussed other ways to treat trace precipitation and left-censored data, but for
convenience, they are simply excluded. The mean wet-day record lengths for point and
areal average precipitation are 7,219 days (equivalent to nearly 20 years) and 14,043 days
(more than 38 years), respectively. The distributions of wet-day record length are shown
in Figure 3. As expected, the proportion of wet-days in the areal average precipitation
data set is higher than that in the point precipitation data set.
[*Figure* 3 *goes here*]

## 275  3. Methodology

This section describes the methods of analysis used for assessing the goodness-of-
fit of various distributional hypotheses, namely, L-moment diagrams and probability plot
correlation coefficients.

### 279  3.1 L-Moment Diagrams

L-moment diagrams are now a widely accepted approach for evaluating the
goodness of fit of alternative distributions to observations. The theory and application of
L-moments introduced by Hosking (1990) is now widely available in the literature
(Stedinger et al., 1993;Hosking and Wallis, 1997), hence it is not reproduced here.
The distribution of daily precipitation totals is highly skewed due to the large
proportion of days with zero precipitation. Higher order conventional moment ratios such
as skewness and kurtosis are very sensitive to extreme values and can exhibit enormous
downward bias even for extremely large sample sizes (Vogel and Fennessey, 1993) as is
the case here. However, L-moment ratios are approximately unbiased in comparison to
conventional moment ratios, thus providing a particularly useful tool for investigating the
pdf of precipitation series.
L-moment ratio diagrams provide a convenient visual way to view the
characteristics of sample data compared to theoretical statistical distributions. The L-
moment diagrams: L-Kurtosis ($\tau_4$) vs L-Skew ($\tau_3$) and L-Cv ($\tau_2$) vs L-Skew ($\tau_3$) enable us
to compare the goodness of fit of a range of three-parameter, two-parameter, and one-
parameter (or special case) distributions. Table 2 displays distributions analyzed by
means of the $\tau_4$ vs $\tau_3$ L-moment ratio diagrams.
[*Table* 2 *goes here*]
Table 3 displays distributions analyzed by means of the $\tau_2$ vs $\tau_3$ L-moment ratio
diagrams.
[*Table* 3 *goes here*]



L-moment ratio diagrams have been used before to examine the distribution of
series of annual maximum precipitation data (Pilon et al., 1991;Park and Jung, 2002;Lee
and Maeng, 2003;Papalexiou and Koutsoyiannis, 2013) and left-censored records
(Deidda and Puliga, 2006). Other than the two recent global studies by Papalexiou and
Koutsoyiannis (2012, 2016) which examined the agreement between empirical and
theoretical relationships between L-Cv and L-Skew, this is the only study we are aware
of, in which a set of uncensored daily precipitation records have been subjected to such a
comprehensive L-moment goodness-of-fit analysis. L-moment estimators were chosen in
this study for a variety of reasons: (1) they are easily computed and nicely summarized
by Hosking and Wallis (1997) for all the cases considered in this study, and (2) estimates
of L-moments unbiased and estimates of their ratios are nearly unbiased, and thus for the
extremely large sample sizes considered here, sampling variability of empirical L-
moment ratios will be extremely small especially when contrasted to distributional choice
comparisons.

### 3.2 Probability plot correlation coefficient goodness-of-fit evaluation

Probability plots are constructed for each of the full record and wet-day series
using L-moment estimators of the distribution parameters (see Hosking and Wallis
(1997)) for the distributions indicated in Table 4.

[*Table* 4 *goes here*]

The goodness of fit of each probability plot is summarized using a probability plot
correlation coefficient (PPCC, or simply, r). The PPCC statistic has a maximum value of
1. The PPCC has been shown to be a powerful statistic for evaluating the goodness-of-fit
of a very wide range of alternative distributional hypotheses (Stedinger et al., 1993) and
for performing hypothesis tests of various two parameter distributional alternatives.

To construct a probability plot and to estimate a probability plot correlation
coefficient, requires estimation of a plotting position. There are two classes of plotting
positions, those that yield unbiased exceedance probabilities and those that yield unbiased
quantile estimates. The Weibull plotting position given by $p=i/(n+1)$ yields an unbiased
estimate of exceedance probability regardless of the underlying distribution (see
(Stedinger et al., 1993)). Alternatively there would be a unique plotting position to use
for each probability distribution, and it is now well known that unbiased plotting
positions for three parameter distributions require an additional parameter to estimate
within the plotting position. For example, Vogel and McMartin (1991) derived an
unbiased plotting position for the P3 distribution which depends upon the skewness of the
distribution, a parameter which adds so much additional uncertainty to the analysis that
led Vogel and McMartin (1991), after considerable analysis, to not recommend its use.
To put all the distributional alternatives on the same footing, we chose to use the Weibull
plotting position for estimation of all PPCC values.

## 4. Results and analysis

### 4.1 L-Moment Diagrams

### 4.1.1 L-Cv vs L-Skew





Figure 4 displays empirical and theoretical distributional relationships between L-Cv and L-Skew for point values of daily precipitation (Figure 4a) and areal average values of daily precipitation (Figure 4b). The various curves represent the theoretical relationship between L-Cv and L-Skew for the distributions indicated. Each plotted point represents the empirical relationship between L-Cv and L-Skew for a single precipitation station or catchment. By comparing the empirically derived points with the theoretical curves, it is possible to see the degree to which the statistical character of the data record matches those of the candidate distributions. We emphasize again, that the sample sizes are large enough in this study so that one may, approximately, ignore sampling variability in all L-moment diagrams. This phenomenon was nicely illustrated in Figure 2 of Blum et al. (2017) for record lengths similar to those used here, but corresponding to daily streamflow records.

The empirical L-moment ratios corresponding to the full-record and wet-day point precipitation series fall within completely different regions in Figure 4a, which is due to the fact that the full-record point precipitation series contain a very large number of zero observations. In contrast, there are much fewer zero observations in the catchment full-record precipitation series thus the empirical L-moment ratios corresponding to the full-record and wet-day catchment precipitation series overlap roughly 50% of the time as shown in Figure 4b.

Both Figure 4a and Figure 4b illustrate a nearly linear relationship between the L-Skew and L-Cv for the two types of full record series. Importantly, these two lines of points, however, do not fall along any of the theoretical curves, demonstrating that the 2-parameter Gamma distribution cannot describe the tail behavior of full-record series of precipitation as has often been assumed in the past.

[*Figure* 4 *goes here*]

In Figure 4a, the wet-day series' points fall primarily within a region bounded by the G2 and GP2 theoretical curves, with the W2 passing through some of the points. In Figure 4b, the wet-day series' points fall primarily in the upper region of the W2 theoretical curve, with the G2 passing through some of the points. These patterns do not indicate a clearly preferred distribution, especially considering that the large sample sizes associated with these series result in negligible sampling variability. Blum et al. (2017, Figure 2) used L-moment diagrams for complete series of daily streamflow observations to demonstrate that the sampling variability in L-moment ratios is negligible for the sample sizes considered in this study. Thus, the scatter shown in Figure 4 is likely due to real distributional differences rather than due to sampling variability as is often the case when one constructs L-moment diagrams for short AMS precipitation records.

## 4.1.2 L-Kurtosis vs L-Skew

Figure 5 displays empirical and theoretical distributional relationships between L-Kurtosis vs L-Skew point values of daily precipitation (Figure 5a) and areal average values of daily precipitation (Figure 5b). The plotted points for the two full record series follow a linear relationship approximately, but the relationships are remarkably similar to the theoretical curve for the Pearson Type-III (P3) distribution. In fact, the P3 pdf seems to be the only 3-parameter distribution that could possibly fit the full record data. It is



worth noting that the overall lower bound of L-Kurtosis for all distributions falls below
but quite close to the P3 curve at high L-Skew values in Figure 5.

[*Figure* 5 *goes here*]

The estimated L-moment ratios of the wet-day series of point precipitation in
Figure 5a reveal more scatter on the plot than for the corresponding full-record series. In
this case, the closest theoretical curve to the wet-day points is also the P3 distribution, but
the fit is less striking for the wet-day series than for the corresponding full record series.
In Figure 5b, the L-moment ratios of the wet-day series of areal average precipitation
shows less scatter than for the corresponding full record series and in this case of areal
rainfall the P3 theoretical curve passes through most of the points for both the full and
wet-day series.  Though the fit of the wet-day series to P3 is less striking than for the full
record series, the L-moment ratio estimates occupy a space that can be well represented
by the Kappa distribution, which occupies not a curve, but a region of the L-Kurtosis vs
L-Skew diagram as shown in Figure A1 of Hosking and Wallis (1997).  See Hosking
(1994) and Hosking and Wallis (1997, Appendix A10) for a complete description of the
4-parameter Kappa distribution.
**4.2 PPCC**
**4.2.1 Standard boxplots of PPCC**

The L-moment diagrams successfully identify two potential candidate
distributions for representing the full-record and wet-day daily precipitation series at the
point and catchment scales.  The PPCC statistic offers another quantitative method for
comparing the goodness of fit of different distributions to the daily precipitation
observations.  Tables 5 and 6 summarize the central tendency and spread of the values of
PPCC for each of the distributions for both the full-record and wet-day series of point and
catchment scale daily precipitation, respectively.  The highest values for the mean,
median, 95[th] percentile, and 5[th] percentile of the PPCC are shown in bold type.  The
lowest values of the sample standard deviation of the PPCC values, denoted ŝ, are also
shown in bold.  Figure 6 illustrates box-plots of the values of PPCC for distributions
fitted to the full-record and wet-day series of daily precipitation data at the point scale.
Figure 7 shows box-plots of PPCC values for distributions fitted to the full-record and
wet-day precipitation series at the catchment scale.

[*Table* 5 *goes here*]

[*Table* 6 *goes here*]

[*Figure* 6 *goes here*]

[*Figure* 7 *goes here*]

Figure 6 and Table 5 indicate that for the full-record series of point daily
precipitation depths, only the G2, P3, and KAP distributions perform well. On the other
hand, for the wet-day series of point daily precipitation, all the distributions have median
PPCCs well above 0.9.  The same situation appears in the areal average precipitation
shown in Figure 7 and Table 6, except that the median PPCCs of the remaining four
distributions for the wet-day series are significantly lower than the corresponding values
for point precipitation.





The insets in Figures 6 and 7 show detailed views of the boxplots of PPCC values
for the G2, P3, and KAP distributions for point and areal average daily precipitation.
Both types of precipitation data shows the same results that the P3 is the best performing
distribution on average for the full-record series, but the KAP distribution shows the
highest PPCCs on average for the wet-day series.

### 4.2.2 Graphical comparison of P3, G2, and KAP

Across all previous comparison, the P3, G2, and KAP are the most likely
distributions for describing daily precipitation at the point or catchment scales. The
insets in Figures 6 and 7 identify the distributions that exhibit the best fit to the observed
series. However, these inserts do not indicate by how much the best performing
distribution outperforms the second or third best. For this purpose, pairwise comparisons
of the PPCC values of two highly performing distributions for all the stations and
catchments are instructive. A simple graphical method can accomplish this goal.
Figure 8a and Figure 8b compare the PPCC values of the P3 (vertical axis) and
G2 (horizontal axis) distributions for the full-record and wet-day series of point daily
precipitation, respectively. Approximately 98% of stations are displayed on both figures;
the remaining stations lie outside the plot domains. Points lying above the diagonal line
indicate that the P3 distribution has a higher PPCC for that particular station, and points
lying below the diagonal line indicate the G2 results in a higher PPCC. The full-record
plot (Figure 8a) shows that in nearly every case, the P3 distribution outperforms the G2
distribution. When the G2 does outperform the P3, the PPCCs are both very high and
nearly equal. The wet-day plot shows that the P3 distribution performs significantly
better than the G2 distribution in many cases. Thus, we conclude the P3 distribution
better represents wet-day daily point precipitation than the more commonly used G2
distribution, in nearly every case.

[*Figure* 8 *goes here*]

Figures 8c and Figure 8d compare the PPCC values of P3 and G2 for the full
record series and wet-day series of areal average precipitation, respectively. The results
are nearly the same as for the point precipitation in the sense that most points are above
the diagonal line; while for a few catchments whenG2 does outperform P3, the points lie
on the dividing line, showing only very slight superiority.
Figure 9a and Figure 9b display similar plots comparing the KAP (vertical axis)
and P3 (horizontal axis) distribution, for the full-record and wet-day series of point
precipitation, respectively. For the full-record series the P3 distribution outperforms the
KAP distribution with most of the points lying below the dividing line; whereas, for the
wet-day series, the KAP distribution outperforms the P3 distribution for a majority of
sites.

[*Figure* 9 *goes here*]

The same conclusion can be obtained for the full-record series of areal average
precipitation in Figures 9c except that the better distribution does not dominate, with only
63% points have higher PPCC for P3 distributions. For the wet-day series of areal
average precipitation in Figures 9d, the performance of KAP distribution is comparable
with that of P3 distribution with almost the same number of points lying in each region.





It is somewhat surprising that the 3-parameter P3 distribution outperforms the 4-
parameter KAP distribution because the extra information contained in the 4th parameter
(essentially a second shape parameter in the case of the Kappa distribution) would be
expected to lead to a better goodness-of-fit. The L-moment diagram (Figure 5), however,
shows that the fit of the full record data to the P3 theoretical curve is so good that a 4th
parameter could be extraneous. Additionally, it should be noted that the pattern of the
full record stations or watersheds on the L-Kurtosis vs L-Skew plot approaches the
overall lower bound for all distributions, a place where the Kappa distribution parameter
estimates may become less accurate. The "h" shape parameter, for example, approaches
infinity in this region (see Hosking and Wallis (1997, Figure A1)).

## 5. Conclusions

This study has demonstrated that L-moment diagrams and probability plot
correlation coefficient goodness of fit evaluations can provide new insight into the
distribution of very long series of daily precipitation at both the point and catchment
scales. Though the commonly used 2-parameter Gamma distribution performs fairly well
on the basis of traditional goodness-of-fit tests, L-moment diagrams and probability plot
correlation coefficient goodness of fit evaluations reveal that very long series of
uncensored daily point and areal average precipitation are better approximated by a
Pearson-III distribution and importantly, they do not resemble any of the other commonly
used distributions.
We conclude that for representing uncensored, full record daily precipitation at
the point and catchment scales, the 3-parameter Pearson-III distribution performs
remarkably well. For cases in which only wet-day precipitation amounts are required, the
Pearson-III distribution is comparable with the 4-parameter Kappa distribution for the
areal average precipitation; when the point precipitation is of concern, the Kappa
distribution should be the distribution of choice. We also conclude that future
investigations should consider comparisons between the generalized Gamma distribution
introduced by Papalexiou and Koutsoyiannis (2012, 2016) for wet-day daily precipitation
and both the Pearson type III and Kappa distributions recommended here.

## Acknowledgements

This work is partially supported by the National Natural Science Foundation of China
(No. 91547116, 51709033, 91647201).

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

**Table captions:**
**Table 1:** Review of literature pertinent to daily precipitation probability distribution
selection.
**Table 2:** Table 2: Theoretical probability distributions presented on the L-Kurtosis vs L-
Skew L-moment diagram. *Italicized distributions are special cases of other distributions*.
**Table 3:** Theoretical probability distributions presented on the L-Cv vs L-Skew L-
moment diagram.
**Table 4:** Distributions used in probability plot goodness of fit evaluations.
**Table 5:** Central tendency and spread of values of PPCC for the 237 precipitation
stations.
**Table 6:** Central tendency and spread of values of PPCC for the 305 areal average
precipitation catchments.
**Figure captions:**
**Figure 1:** Map showing locations of a) 237 precipitation gaging stations, and b) 305
catchments.
**Figure 2:** Distribution of full record length of point precipitation base on weather stations.





**Figure 3:** Distribution of wet-day record length: a) point precipitation; and b) areal average precipitation over watersheds. Days with zero precipitation are removed in the wet-day records

**Figure 4:** L-Cv vs L-Skew L-moment ratio diagram of sample L-moments and theoretical distributions: a) point precipitation; and b) areal average precipitation depths.

**Figure 5:** L-Skew vs L-Kurtosis L-moment ratio diagram of sample L-moments and theoretical distributions: a) point precipitation; and b) areal average precipitation depths. Logistic (L), Normal (N), Uniform (U), Gumbel (G), and Exponential (E) distributions appear as a single point.

**Figure 6:** Standard boxplots of r for all 7 distributions evaluated for a) full record, and b) wet-day series of point precipitation depths.

**Figure 7:** Standard boxplots of r for all 7 distributions evaluated for a) full- record, and b) wet-day series of areal average precipitation depths.

**Figure 8:** Comparison of PPCC (r) values for the P3 (vertical axis) and G2 (horizontal axis) distributions for the a) point precipitation depths' full -record, b) point precipitation depths' wet-day, c) areal average precipitation depths' full-record, and d) areal average precipitation depths' wet-day series. Points lying above the line represent stations with a higher r for the P3 distribution than G2 distribution.

**Figure 9:** Comparison of r values for P3 (horizontal axis) and KAP (vertical axis) distributions for the a) point precipitation depths' full-record, b) point precipitation depths' wet-day, c) areal average precipitation depths' full-record, and d) areal average precipitation depths' wet-day series.

## Tables



**Table 1:** Review of literature pertinent to daily precipitation probability distribution selection.

*1. Stochastic Precipitation Modelling:*

| Author | Year | Stations | Series type | Duration | Distribution | Justification |
|---|---|---|---|---|---|---|
| Thom | 1951 | | Wet-day | 1-day | Gamma | |
| Buishand | 1978 | 6 | Wet-day | 1-day | Gamma | Cv-Cs ratio |
| Geng et al | 1986 | 6 | Wet-day, by month | 1-day, monthly | Gamma | Regress. fit: β vs mean wet-day depth |
| Woolhiser and Roldan | 1982 | | Wet-day | 1-day | Mixed Exponential | MLE, Akaike Information Criterion |
| Duan et al | 1995 | 1 | Wet-day, by month | 1-day | Calib. W2, Gamma | MLE, Chi-sq test |
| Wilks | 1998 | 25 | Wet-day | 1-day | Mixed Exponential | MLE, goodness of fit |
| Waterson and Dix | 2003 | | Wet-day | 1-day | Gamma | Literature |
| Burgueno et al | 2005 | 75 | Wet-day | 1-day | Exponential, Weibull | Normalized Rainfall Curve |
| Waterson | 2005 | | Wet-day | 1-day | Gamma | Literature |

*2. Precipitation Frequency Analysis*

| Author | Year | Stations | Series type | Duration | Distribution | Justification |
|---|---|---|---|---|---|---|
| Hershfield  (TP-40) | 1962 | | AMS | 24 hour | Gumbel | |
| Pilon et al | 1991 | 75 | AMS | 5 min - 24 hour | GEV | L-moments |
| Naghavi & Yu | 1995 | 25 | AMS | 1-24 hour | GEV | L-moments, PWMs, Monte Carlo experiments |
| Park and Jung | 2002 | 61 | AMS | 1, 2-day | Kappa(4) | L-moments |
| Lee and Maeng | 2003 | 38 | AMS | 1-day | GEV, GLO | L-moments |
| Bonnin et al | 2006 | | AMS | 5 min - 24 hour | GEV | L-moments |
| Shoji and Kitaura | 2006 | 243 | Complete, Wet-day | Hour, Day, Month, Year | Lognormal, Weibull | Goodness of fit |
| Deidda and Puliga | 2006 | 200 | Left Censored Wet-day PDS | 1-day | Generalized Pareto | "Failure-to-reject" method, L-moments |
| Papalexiou and Koutsoyiannis | 2012 | 11,519 | Wet-Day | 1-day | Generalized Gamma | L-moments |
| Papalexiou and Koutsoyiannis | 2013 | 15,137 | AMS | 1-day | GEV | L-moments |
| Papalexiou and Koutsoyiannis, | 2016 | 14,157 | Wet-Day, by month | 1-day | Generalized Gamma and Burr type XII | L-moments and Goodness-of-fit |

*3. Precipitation Trends and Climate Change*

| Author | Year | Stations | Series type | Duration | Distribution | Justification |
|---|---|---|---|---|---|---|
| Waggoner | 1989 | 55 | Monthly | 1-month | Gamma | Literature Review |
| Groisman et al | 1999 | 1313 | Summer (wet-day) | 1-day | Gamma | Literature Review, goodness of fit to |





| | | | | | | extreme rainfall quantiles |
|---|---|---|---|---|---|---|
| Wilby and Wigley | 2002 | GCM | Seasonal | 1-day | Gamma | Literature Review |
| Yoo et al | 2005 | 31 | Monthly (wet-day) | 1-day | Gamma | Literature Review |
| Watterson | 2005 | GCM | January, July | 1-month (daily forced) | Gamma | Literature Review |





**Table 2:** Theoretical probability distributions presented on the L-Kurtosis vs L-Skew L-moment diagram. *Italicized distributions are special cases of other distributions.*

| Distribution | Abbreviation | Parameters |
|---|---|---|
| Generalized Extreme Value Type III | GEV | 3 |
| Generalized Logistic | GLO | 3 |
| Generalized Pareto | GPA | 3 |
| Lognormal | LN3 | 3 |
| Pearson Type III | P3 | 3 |
| *Exponential* | E | 2 |
| *Gumbel* | G | 2 |
| *Normal* | N | 2 |
| *Logistic* | L | 2 |
| *Uniform* | U | 1 |

**Table 3:** Theoretical probability distributions presented on the L-Cv vs L-Skew L-moment diagram.

| Distribution | Abbreviation | Parameters |
|---|---|---|
| Gamma | G2 | 2 |
| Generalized Pareto | GP2 | 2 |
| Lognormal | LN2 | 2 |
| Weibull | W2 | 2 |

**Table 4:** Distributions used in probability plot goodness of fit evaluations.

| Distribution | Abbreviation | Parameters |
|---|---|---|
| Generalized Extreme Value Type III | GEV | 3 |
| Generalized Logistic | GLO | 3 |
| Generalized Pareto | GPA | 3 |
| Lognormal | LN3 | 3 |
| Pearson Type III | P3 | 3 |
| Gamma | G2 | 2 |
| Kappa | KAP | 4 |

**Table 5:** Central tendency and spread of values of PPCC for the 237 precipitation stations.

| Distribution | Full Record | | | *Percentiles* | | Wet Day | | | *Percentiles* | |
|---|---|---|---|---|---|---|---|---|---|---|
| | Mean | Median | ŝ | 95th | 5th | Mean | Median | ŝ | 95th | 5th |
| P3 | **0.9953** | **0.9962** | **0.0045** | **0.9991** | **0.9892** | 0.9952 | 0.9971 | 0.0063 | 0.9995 | 0.9872 |
| GEV | 0.5949 | 0.5928 | 0.0527 | 0.6755 | 0.5166 | 0.9338 | 0.9375 | 0.0222 | 0.9609 | 0.8944 |
| GPA | 0.6192 | 0.6177 | 0.0604 | 0.7145 | 0.5339 | 0.9793 | 0.9828 | 0.0145 | 0.9949 | 0.9500 |
| GLO | 0.5939 | 0.5922 | 0.0509 | 0.6708 | 0.5172 | 0.9115 | 0.9154 | 0.0235 | 0.9423 | 0.8734 |
| LN3 | 0.7975 | 0.8078 | 0.0545 | 0.8731 | 0.7055 | 0.9838 | 0.9855 | 0.0075 | 0.9924 | 0.9727 |
| G2 | 0.9945 | 0.9954 | 0.0046 | 0.9988 | 0.9876 | 0.9925 | 0.9949 | 0.0079 | 0.9990 | 0.9789 |
| KAP | 0.9780 | 0.9784 | 0.0137 | 0.9926 | 0.9644 | **0.9971** | **0.9985** | **0.0048** | **0.9997** | **0.9915** |



**Table 6:** Central tendency and spread of values of PPCC for the 305 areal average precipitation catchments.

| Distribution | Full Record | | | Percentiles | | Wet Day | | | Percentiles | |
|---|---|---|---|---|---|---|---|---|---|---|
| | Mean | Median | § | 95th | 5th | Mean | Median | § | 95th | 5th |
| P3 | **0.9972** | **0.9975** | **0.0023** | 0.9993 | **0.9941** | **0.9977** | 0.9985 | **0.0028** | 0.9996 | **0.9936** |
| GEV | 0.6757 | 0.6706 | 0.0666 | 0.8014 | 0.5836 | 0.8003 | 0.7965 | 0.0474 | 0.8917 | 0.7264 |
| GPA | 0.7247 | 0.7177 | 0.0795 | 0.8711 | 0.6140 | 0.8688 | 0.8687 | 0.0484 | 0.9586 | 0.7894 |
| GLO | 0.6654 | 0.6607 | 0.0608 | 0.7772 | 0.5803 | 0.7800 | 0.7750 | 0.0441 | 0.8669 | 0.7101 |
| LN3 | 0.8717 | 0.8736 | 0.0444 | 0.9409 | 0.8035 | 0.9362 | 0.9373 | 0.0224 | 0.9737 | 0.8983 |
| G2 | 0.9967 | 0.9971 | 0.0024 | 0.9992 | 0.9935 | 0.9974 | 0.9985 | 0.0034 | 0.9996 | 0.9924 |
| KAP | 0.9959 | 0.9968 | 0.0034 | **0.9996** | 0.9898 | 0.9976 | **0.9987** | 0.0026 | **0.9998** | 0.9929 |





# Figures

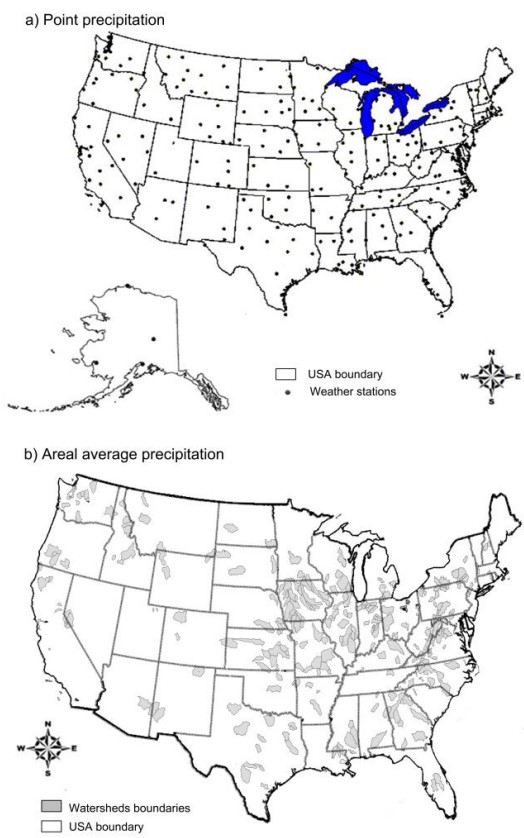

**Figure 1:** Map showing locations of a) 237 point precipitation gaging stations, and b) 305 MOPEX catchments.

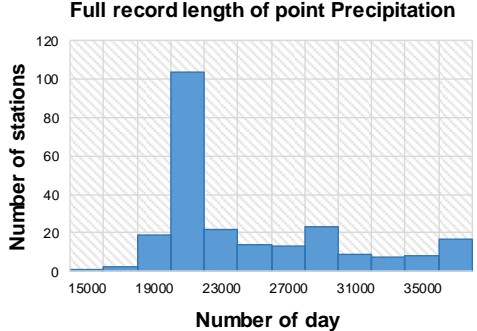

**Figure 2:** Distribution of length of records of point daily precipitation data for the 237 gaging stations depicted in Figure 1a.





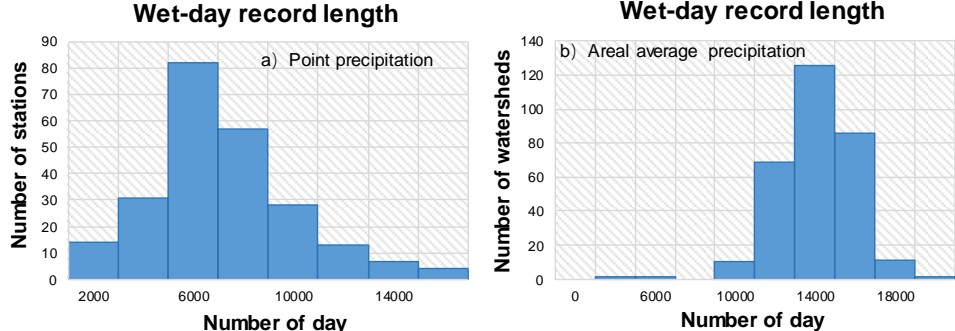

**Figure 3:** Distribution of wet-day record lengths corresponding to the two datasets: a) point precipitation; and b) areal average precipitation over catchments. Days with zero precipitation are removed in the wet-day records

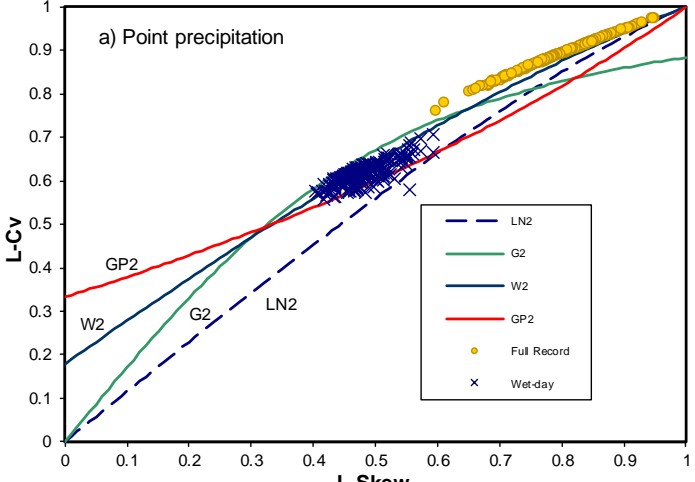



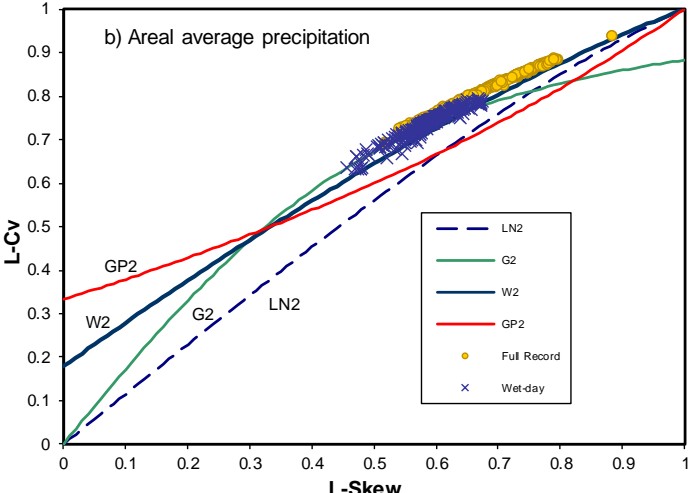

**Figure 4:** L-Cv vs L-Skew L-moment ratio diagram of sample L-moments and theoretical distributions for: a) point daily precipitation; and b) areal average daily precipitation depths.

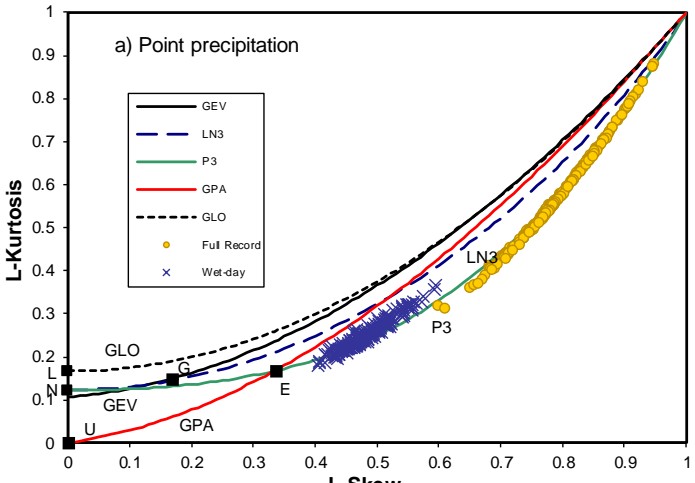





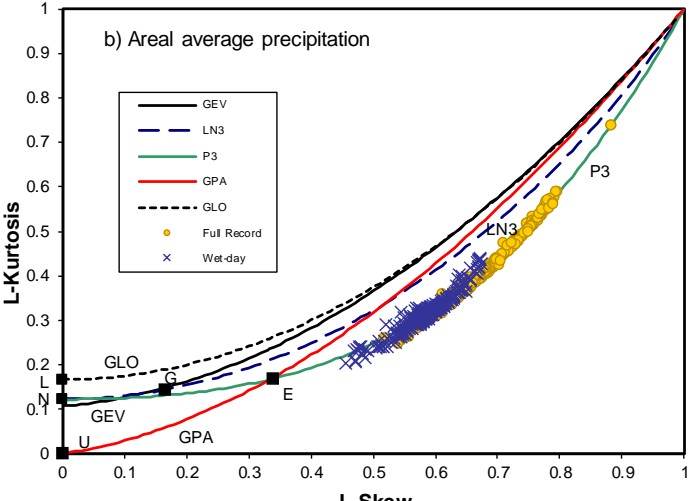

**Figure 5:** L-Skew vs L-Kurtosis L-moment ratio diagram of sample L-moments and theoretical distributions for: a) point daily precipitation; and b) areal average daily precipitation depths. Note that Logistic (L), Normal (N), Uniform (U), Gumbel (G), and Exponential (E) distributions appear as a single point.





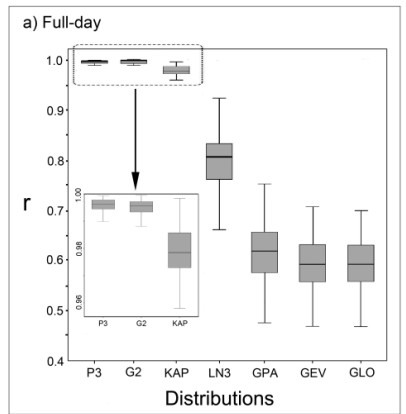
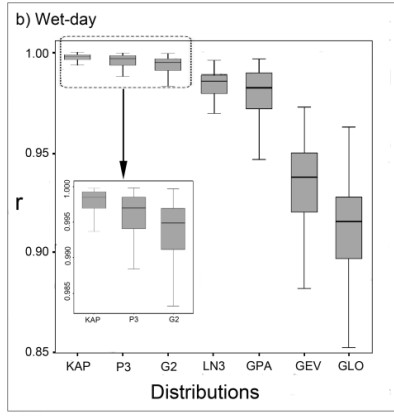

**Figure 6:** Standard boxplots of r for all 7 distributions evaluated for a) full record, and b) wet-day series of point precipitation depths.



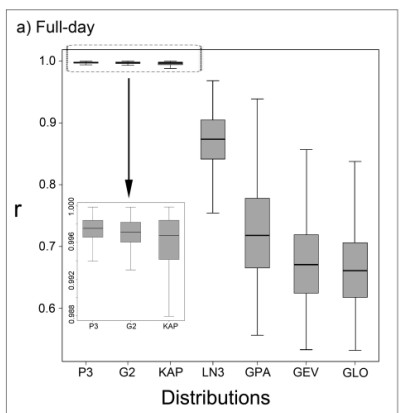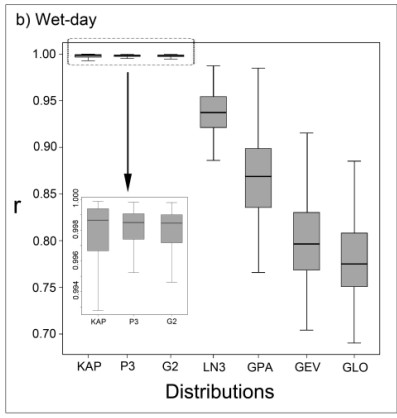

**Figure 7:** Standard boxplots of r for all 7 distributions evaluated for a) full- record, and b) wet-day series of areal average precipitation depths.





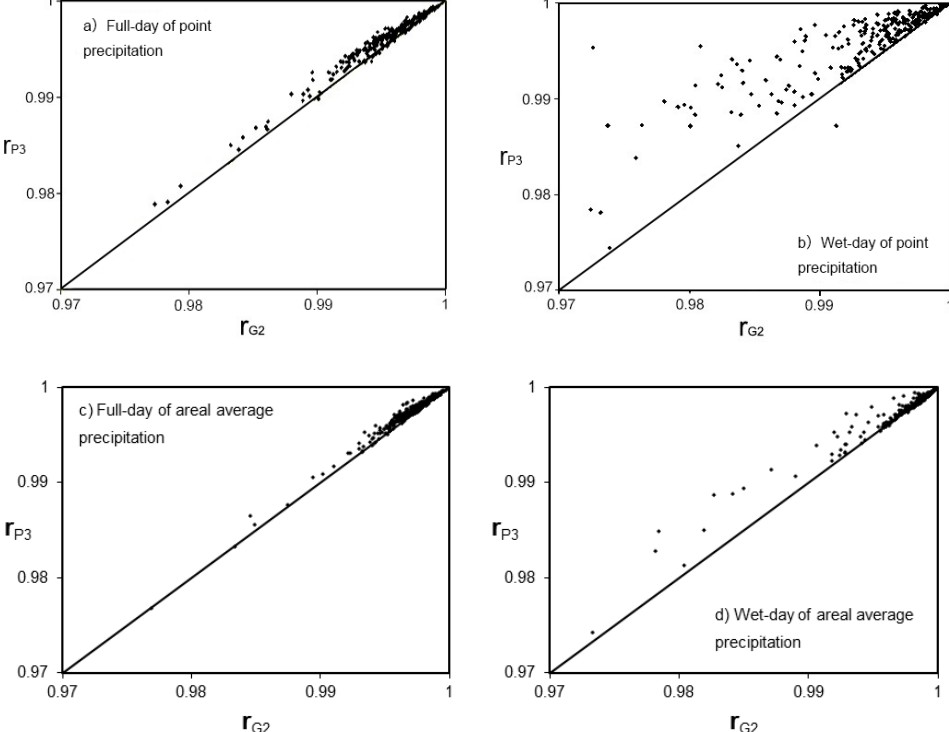

**Figure 8:** Comparison of PPCC (r) values for the P3 (vertical axis) and G2 (horizontal axis) distributions for the a) point precipitation depths' full-record, b) point precipitation depths' wet-day, c) areal average precipitation depths' full-record, and d) areal average precipitation depths' wet-day series. Points lying above the line represent stations with a higher r for the P3 distribution than G2 distribution.





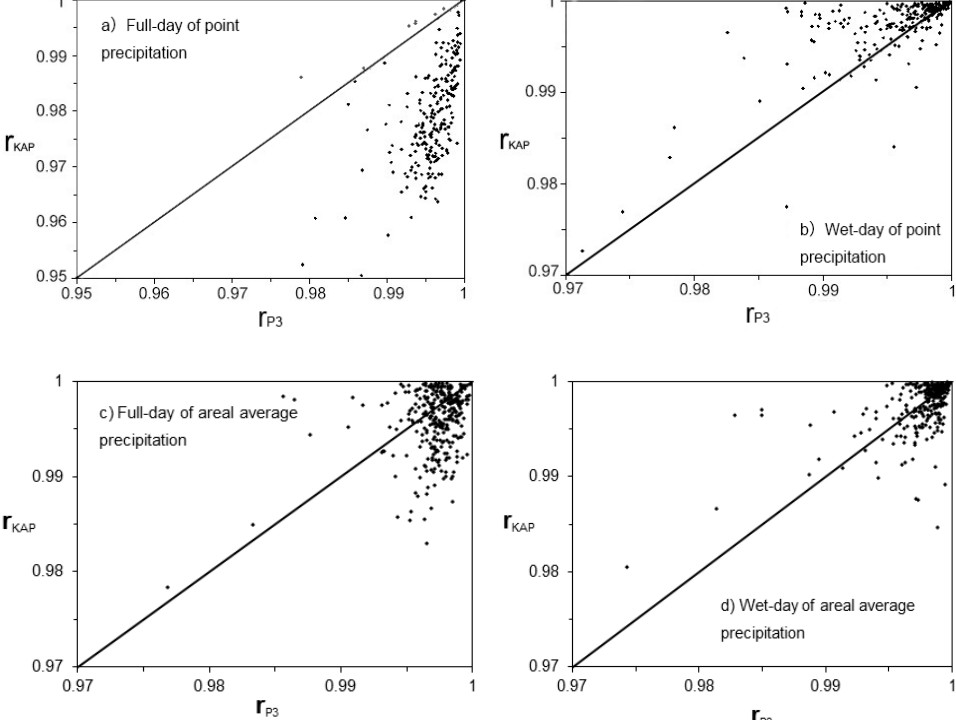

**Figure 9:** Comparison of r values for P3 (horizontal axis) and KAP (vertical axis) distributions for the a) point precipitation depths' full-record, b) point precipitation depths' wet-day, c) areal average precipitation depths' full-record, and d) areal average precipitation depths' wet-day series.