# Peer review of "The Probability Distribution of Daily Precipitation at the Point 1 and Catchment Scales in the United States 2"

_Hydrology and Earth System Sciences, 2018_

## Referee Comment (RC1) · S.M. Papalexiou (Referee) · 28 Mar 2018

**Data from processes with mixed-type marginals cannot be treated using continuous marginals**

Simon Michael Papalexiou

Department of Civil and Environmental Engineering, University of California, Irvine, CA, USA
(simon@uci.edu)

The  paper of (Ye et al., 2018) entitled "*The Probability Distribution of Daily Precipitation at the Point and Catchment Scales in the United States*" deals with an important topic, i.e., the identification of probability distributions to describe the daily rainfall both at station level but also at large catchments. The paper has a nice and clear logical structure, it is easy to read (quite a rare quality) and it is the first study as far as I know that deals with a large number of records at the catchment level. Clearly, there is potential in this study, but unfortunately in my opinion there is a fundament issue that needs to be addressed, i.e., the part that uses the whole record of precipitation values including zeros.

**Apples with oranges**

It is well-known that many processes in nature, including precipitation, are intermittent processes, and therefore their marginal distribution is of mixed-type, i.e., it has both probability mass (pmf) to express concentration at zero and probability density (pdf) to express the nonzero values. Of course the expressions of the distribution function $F_X(x)$, the pdf $f_X(x)$ and the quantile function $Q_X(u)$ can be related to the conditional expressions for $X|X > 0$. Thus, if $p_0$ is the probability dry, then the cdf, pdf (it is not actually pdf, it is pmf and pdf at the same time: dirac delta notation can be used to unify to pdf) and quantile functions of $X$ are given by

$$F_X(x) = (1 - p_0)F_{X|X>0}(x) + p_0 \quad x \geq 0 \tag{1}$$

$$f_X(x) = \begin{cases} p_0 & x = 0 \\ (1 - p_0)f_{X|X>0}(x) & x > 0 \end{cases} \tag{2}$$

$$x_u = Q_X(u) = \begin{cases} 0 & 0 \leq u \leq p_0 \\ Q_{X|X>0}\left(\dfrac{u - p_0}{1 - p_0}\right) & p_0 < u \leq 1 \end{cases} \tag{3}$$

Now, this affects profoundly the expressions of moments, as the $q$-th raw moment is given by

$$m(q) = (1 - p_0) \int_0^\infty x^q f_{X|X>0}(x) \mathrm{d}x = (1 - p_0) m_{X|X>0}(q) \tag{4}$$

27 and of course using the well-known formulas that relate the central moments to raw
28 moments we can find easily the expressions of the mean, variance, skewness, kurtosis etc.
29 For example, the mean, variance, and the third and fourth central moments are given by

$$\mu_X = (1 - p_0)\mu_{X|X>0} \tag{5}$$

$$\sigma_X^2 = (1 - p_0)\sigma_{X|X>0}^2 + p_0(1 - p_0)\mu_{X|X>0}^2 \tag{6}$$

$$\mu(3) = 2m(1)^3 - 3m(1)m(2) + m(3) \tag{7}$$

$$\mu(4) = -3m(1)^4 + 6m(1)^2 m(2) - 4m(1)m(3) + m(4) \tag{8}$$

30 where of course the raw moments in Eqs (7)-(8) should be replaced using Eq (4).
31      I show these expressions using product moments as they are analytical to stress how
32 summary statistics are affected by the presence of zeros. For example, if product moment
33 ratio-plots were used to identify an appropriate distribution, using empirical statistics of the
34 whole record would be valid only if compared with the corresponding theoretical curves that
35 express the mixed-type distribution.
36      The situation with L-moments is the same. Particularly, we can define the L-moments
37 for the mixed-type marginal, if I am not mistaken, as

$$\lambda_1 = \int_{p_0}^1 Q_{X|X>0}\left(\frac{u - p_0}{1 - p_0}\right) \mathrm{d}u \tag{9}$$

$$\lambda_2 = \int_{p_0}^1 Q_{X|X>0}\left(\frac{u - p_0}{1 - p_0}\right)(2u - 1)\mathrm{d}u \tag{10}$$

$$\lambda_3 = \int_{p_0}^1 Q_{X|X>0}\left(\frac{u - p_0}{1 - p_0}\right)(6u^2 - 6u + 1)\mathrm{d}u \tag{11}$$

$$\lambda_4 = \int_{p_0}^1 Q_{X|X>0}\left(\frac{u - p_0}{1 - p_0}\right)(20u^3 - 30u^2 + 12u - 1)\mathrm{d}u \tag{12}$$

38 for which analytical expressions can be derived for some specific distributions.

The authors here are presenting in their L-ratios plot a comparison of summary statistics estimated from the whole record (mixed-type data) with the theoretical curves or points of the continuous distributions and not of the mixed-type distributions which can be derived from the equation I previously presented.

It should be apples with apples and oranges with oranges. Thus, if the authors want to use the whole record they have to construct the corresponding curves for the mixed-type cases. So, the fact that the P3 seems a good choice for the whole records it is just an artifact, as well as the nice and neat concertation of points. It is the changes in probability dry that dominate the statistics. And since the domination comes from the probability dry I would guess that if the authors construct the corresponding curves (for fixed $p_0$; otherwise they form an area) for the mixed-type case they will find that for high $p_0$ values these curves for different conditional distributions are very similar.

This can be easily also verified by empirical points using simulations. In the Fig. 1 I generated synthetic precipitation having the same correlation structure, the same probability dry, i.e., 90%, but two very different marginals (for a method on how to generate precipitation with any marginal, and any correlation structure and preserving intermittency see Papalexiou (2018)). In Fig.1a is precipitation from a Pareto II with tail exponent 0.2 and in Fig. 1b is from an exponential (light tail). One hundred samples were generated for each case and the L-ratio points were estimated (red and blue dots correspond to Pareto and Exponential cases, respectively). As we see in Fig 1.c the L-ratios for the whole sample (including zeros) are essentially the same for the two distributions forming a linear line (note the narrow range, e.g., in skewness from 0.87 to 0.93 and the huge overlapping). On the other hand, the L-ratios in Fig.1d referring only to the nonzero sample they are quite different (see the large range and insignificant overlapping).

[Figure]

**Figure 1:** Synthetic precipitation having the same autocorrelation structure and probability dry (90%) but different marginal distributions, that is (a) Pareto II and (b) Exponential. Sample L-ratio points for one hundred generated samples from each case for the whole samples (c) and the nonzero samples (d).

Thus all parts that refer to the whole record as well as the conclusions drawn from the comparisons with the nonzero samples have to be modified in my opinion.

**Other issues**

1. Lines 363-365: "*demonstrating that the parameter Gamma distribution cannot describe the tail behavior of full-record series of precipitation as has often been assumed in the past.*"

    These lines are just the opportunity for commenting on tail issues. Summary shape statistics are of course affected by the tail behaviour but they are not sufficient to reveal in a robust way the behaviour of the tail if the whole sample is used (I mean all nonzero values) and not values that belong to the tail. For example  in the paper the authors cite (Papalexiou and Koutsoyiannis, 2016) after the fitting using L-moments various

measures were proposed in order to compare the fitting in the most extreme value, the largest extremes the whole sample etc. The author can see that the performance of distributions changed, still the GG performed better but the BrXII increased its performance too. I just want to say that indeed this approach can favour specific distributions and exclude others like the G2 the authors mention, yet this is based judging the whole distributional shape properties and it is not really robust to judge on the tail when using the whole nonzero sample. Also other global studies indicated the sub exponential nature of tails focusing on using only "tail" data (Papalexiou et al., 2013; Serinaldi and Kilsby, 2014); the latter was also applied in a seasonal basis, which by the way might be also a nice idea, i.e., the authors to explore seasonal variation.

2. The P3 distribution is just the two-parameter Gamma distribution (G2) with an additional location parameter which does not affect the shape characteristics and thus $\tau_3$ and $\tau_4$. So the curve of P3 shown in $\tau_4 - \tau_3$ ratio plots is the same as the G2. And obviously they have the same tail. The same holds for GPA and GP2 and for any other distribution that has one shape parameter and additional location parameters are added. Maybe to ease the reader, as different formulations can be found in the literature, it would be no harm to add a table of the distributions functions used.

3. The Weibull distribution could also be added in the analysis as a representative of distributions with stretched exponential tails.

4. When we use distributions with a location parameter to describe a positive variable like the nonzero precipitation this parameter might end far from zero or even negative indicating a lower bound. So, this distribution cannot be used in stochastic modelling of precipitation as it will result in inconsistent values. It would be interesting the authors to actually show some box plots of the estimated parameters.

5. The principle of parsimony should always be applied. If the authors, generate samples from a 4-parameter distribution like the kappa and try to estimate a posteriori the parameters, even for the sample sizes used here, they will find a huge variability that makes, in my opinion, the operational use of 4-parameter distributions quite risky. Of course a 4-parameter distribution like the kappa has a great flexibility, yet this does imply that a better fitting to an observed sample is a better choice to extrapolate values for large return periods.

6. The authors, since this is the first large scale study on catchment precipitation, could provide some analysis on the relation of catchment size and distributional shape. As the spatial averaging process will tend to make the distributions more bell-shaped and with thinner tails. This is the explanation of the performance decrease of the heavy-tailed distribution shown in Fig. 7 compared to Fig. 6 (commenting on the Wet-day; full-day results should be modified).

7. Also, some regions of the USA, mainly in Midwest, show quite intense changes (or maybe natural variability) on extremes. The authors could also comment on that or do a quick extra analysis on the daily precipitation.

8. Finally, I believe the literature should be updated with many other works, e.g., there are several papers that are using other distributions for daily precipitation, e.g., one that came to mind is the by Wilson and Toumi (2005).

**References**

Papalexiou, S.M., 2018. Unified theory for stochastic modelling of hydroclimatic processes: Preserving marginal distributions, correlation structures, and intermittency. Advances in Water Resources. https://doi.org/10.1016/j.advwatres.2018.02.013

Papalexiou, S.M., Koutsoyiannis, D., 2016. A global survey on the seasonal variation of the marginal distribution of daily precipitation. Advances in Water Resources 94, 131–145. https://doi.org/10.1016/j.advwatres.2016.05.005

Papalexiou, S.M., Koutsoyiannis, D., Makropoulos, C., 2013. How extreme is extreme? An assessment of daily rainfall distribution tails. Hydrol. Earth Syst. Sci. 17, 851–862. https://doi.org/10.5194/hess-17-851-2013

Serinaldi, F., Kilsby, C.G., 2014. Rainfall extremes: Toward reconciliation after the battle of distributions. Water Resour. Res. 50, 336–352. https://doi.org/10.1002/2013WR014211

Ye, L., Hanson, L.S., Ding, P., Wang, D., Vogel, R.M., 2018. The Probability Distribution of Daily Precipitation at the Point and Catchment Scales in the United States. Hydrol. Earth Syst. Sci. Discuss. 2018, 1–28. https://doi.org/10.5194/hess-2018-85

Wilson, P.S., Toumi, R., 2005. A fundamental probability distribution for heavy rainfall. Geophys. Res. Lett. 32, L14812. https://doi.org/10.1029/2005GL022465

---

## Referee Comment (RC2) · Anonymous Referee #2 · 26 Apr 2018

The topic is of interest for the HESS readership and the paper is overall reasonably well written. Unfortunately, the authors have not taken the opportunity to respond to the critique by the other reviewer. The critique seems legitimate and is serious as it suggests that all the full-record results and conclusions are invalid. In addition, the structure of the paper could use some work and the paper seems a bit unfinished as I will explain below.

It should probably be explained in the Introduction why "Establishing a probability distribution that provides a good fit to daily precipitation depths has long been a topic interest".

[Figure]

The research objectives are included in the subsection "Precipitation trends and changes", which isn't really logical. Consider restructuring the Introduction, for example, by adding a "Research objectives" subsection.

The Introduction is almost half of the paper. Considering shortening it or moving the less essential material to a background subsection.

Line 267: Regarding "less than "0.01" recordable precipitation," what are the units of the 0.01? Isn't this threshold too low given the detection limit of gauges (approximately 0.25 mm)?

Can you show some maps of the results to reveal what the spatial patterns in the results look like? Are there any striking differences between, for example, the temperate southeastern and arid southwestern US?

A Discussion section is missing from the paper. What is the broader significance of the results? Are the results representative of the rest of the world?

---

## Referee Comment (RC3) · Anonymous Referee #3 · 26 Apr 2018

General comments:

The topic of this paper is of interest to the HESS readers and has the potential to add to the large body of research on this important topic. However I, too, have some concerns that the authors have not yet responded to the thorough and thoughtful review from Referee 1. In addition to the comments from Referees 1 and 2, I have additional comments related to the methodology and the presentation quality of the paper. These additional comments are described below.

Specific/technical comments:

The 'Introduction' section describes in great detail the vast literature related to the topics of (1) stochastic precipitation modeling, (2) precipitation frequency analysis, and (3) precipitation tends and climate changes. In this thorough review it is apparent that the Pearson Type III (P3) distribution has not been considered as a candidate distribution to describe wet-day, AMS or PDS daily precipitation series. Yet the consideration of the P3 distribution is largely explored in this paper. Recommend the authors add why they believe the P3 is an appropriate distribution for the extreme values of rainfall.

Similar to Referee #2, I believe too much detail is presented in the 'Introduction' section. The lengthy discussion doesn't add to the flow of the paper. Recommend reducing the literature review discussion, highlighting the important studies related to the topics in this paper and refer the reader to Table 1 for a more thorough review of previous studies.

Similar to Referee #2, a 'Discussion' section is missing in this paper and I recommend it be added.

234-239 is interpretive and describes the findings of this paper. This should be moved to the 'Discussion' and/or 'Conclusions' sections.

Similarly, the last sentence in the 'Introduction' section (lines 243-245) is interpretive and should be moved to the 'Discussion' and/or 'Conclusions' sections.

---

## Author Comment (AC1) · 21 Jun 2018

**"The Probability Distribution of Daily Precipitation at the Point and**

**Catchment Scales in the United States" by Lei Ye et al.**

**Response to Simon M. Papalexiou (Referee #1)**

We greatly appreciate you for your constructive comments and suggestions. Our responses to the comments are listed below.

**Comment 1: It only makes sense to advance a single continuous pdf for the wet**

**day case, regardless of where the data arises from, if one adds zeros, the**

**Lmoments will always land exactly on the Pearson Type III curve**

*Response:*

Thank you for pointing this out.   In the revised manuscript, we will eliminate all the results relating to the 'all day' or X>=0 conditions, focusing only on the probability distribution of wet-day precipitation.

**Comment 2: Lines 363-365: "demonstrating that the parameter Gamma**

**distribution cannot describe the tail behavior of full-record series of**

**precipitation as has often been assumed in the past." These lines are just the**

**opportunity for commenting on tail issues. Summary shape statistics are of**

**course affected by the tail behaviour but they are not sufficient to reveal in a**

**robust way the behaviour of the tail if the whole sample is used (I mean all**

**nonzero values) and not values that belong to the tail. For example in the**

**paper the authors cite (Papalexiou and Koutsoyiannis, 2016) after the fitting**

**using L-moments various measures were proposed in order to compare the**

**fitting in the most extreme value, the largest extremes the whole sample etc.**

**The author can see that the performance of distributions changed, still the**

**GG performed better but the BrXII increased its performance too. I just want**

**to say that indeed this approach can favour specific distributions and exclude**

**others like the G2 the authors mention, yet this is based judging the whole**

**distributional shape properties and it is not really robust to judge on the tail when using the whole nonzero sample. Also other global studies indicated the sub exponential nature of tails focusing on using only "tail" data (Papalexiou et al., 2013; Serinaldi and Kilsby, 2014); the latter was also applied in a seasonal basis, which by the way might be also a nice idea, i.e., the authors to explore seasonal variation.**

*Response:*

Thank you for the comments. The sentence will be removed in the revised manuscript since our analysis of the full-record series of precipitation will be eliminated. The reviewer has suggested a good idea to explore the seasonal variation of the distribution of daily rainfall, which will be a portion of our future work. We will consider ideas introduced by Papalexiou and Koutsoyiannis (2016) and others concerning the impact of seasons on rainfall distributions. We are aware that the choice of a suitable distribution for modeling rainfall would be quite different if we were to focus our attention on the extreme tail behavior, as is the case for example, when one fits a distribution to the series of annual maxima or peaks above some threshold. However given our interest in all wet-day daily precipitation, most situations of practical relevance and concern are not with extreme rainfall, thus our attention will focus on the complete series of wet-day amounts, without special attention given to the largest values.

**Comment 3: The P3 distribution is just the two-parameter Gamma distribution (G2) with an additional location parameter which does not affect the shape characteristics and thus $\tau_3$ and $\tau_4$. So the curve of P3 shown in $\tau_4-\tau_3$ ratio plots is the same as the G2. And obviously they have the same tail. The same holds for GPA and GP2 and for any other distribution that has one shape parameter and additional location parameters are added. Maybe to ease the reader, as different formulations can be found in the literature, it would be no harm to add a table of the distributions functions used.**

*Response:*

Thank you. We will add a table to show the distribution functions. We appreciate the comments of the reviewer reminding us of the fact that addition of a location parameter does not impact the shape of the distribution.

**Comment 4: The Weibull distribution could also be added in the analysis as a**

**representative of distributions with stretched exponential tails.**

*Response:*

Thanks. We will add the analysis of Weibull distribution in the revision. Again, we emphasize that our interest is in the entire distribution, without focusing attention on the extreme tail behavior. We leave it to others to evaluate the distribution of annual maximum precipitation to focus attention on extreme tail behavior of daily rainfall amounts.

**Comment 5: When we use distributions with a location parameter to describe a**

**positive variable like the nonzero precipitation this parameter might end far**

**from zero or even negative indicating a lower bound. So, this distribution**

**cannot be used in stochastic modelling of precipitation as it will result in**

**inconsistent values. It would be interesting the authors to actually show some**

**box plots of the estimated parameters.**

*Response:*

We indeed found that some parameters are far from zero too. Box plots will be added to illustrate the behavior of the distribution parameters of interest.

**Comment 6: The principle of parsimony should always be applied. If the authors,**

**generate samples from a 4-parameter distribution like the kappa and try to**

**estimate a posteriori the parameters, even for the sample sizes used here, they**

**will find a huge variability that makes, in my opinion, the operational use of**

**4-parameter distributions quite risky. Of course a 4-parameter distribution**

**like the kappa has a great flexibility, yet this does imply that a better fitting**

**to an observed sample is a better choice to extrapolate values for large return**

**periods.**

*Response:*

We fully agree that in most applications in hydrology, the principle of parsimony is absolutely paramount, due to the short samples available for fitting distribuitons.

However, in this application, with samples sizes in the tens of thousands, concerns over parsimony are not nearly as critical, even when estimating the Kappa distribution. This fact has been shown nicely in the recent paper by Blum et al (2017) where they demonstrated, for similarly length samples of daily streamflow the sampling properties of estimated Lmoments from synthetic samples in their Figure 2.

**Comment 7: The authors, since this is the first large scale study on catchment**

**precipitation, could provide some analysis on the relation of catchment size**

**and distributional shape. As the spatial averaging process will tend to make**

**the distributions more bell-shaped and with thinner tails. This is the**

**explanation of the performance decrease of the heavy-tailed distribution**

**shown in Fig. 7 compared to Fig. 6 (commenting on the Wet-day; full-day**

**results should be modified).**

*Response:*

Thank you for your suggestion. In the revised manuscript, we will explore the relation of catchment size and distribution shape. Of interest is the impact of catchment size on both the LSkewness and LKurtosis of the fitted distributions of wet-day precipitation amounts.

**Comment 8: Some regions of the USA, mainly in Midwest, show quite intense**

**changes (or maybe natural variability) on extremes. The authors could also**

**comment on that or do a quick extra analysis on the daily precipitation.**

*Response:*

Thank you. We will explore/comment on the changes of extremes in Midwest if possible, but again, we remind the reviewer, that our attention is not on the behavior of extreme precipitation, but rather, is on the distribution of the complete series of wet day daily amounts under all conditions, not just extreme conditions.

**Comment 9: I believe the literature should be updated with many other works, e.g.,**

**there are several papers that are using other distributions for daily**

**precipitation, e.g., one that came to mind is the by Wilson and Toumi (2005).**

*Response:*

Thank you. We will update the literature review with a focus on the distribution of wet- day precipitation amounts.

---

## Author Comment (AC2) · 21 Jun 2018

**"The Probability Distribution of Daily Precipitation at the Point and Catchment Scales in the United States" by Lei Ye et al.**

**Response to Referee #2**

We greatly appreciate you for your constructive comments and suggestions. Our responses to the comments are listed below.

**Comment 1: It should probably be explained in the Introduction why "Establishing a probability distribution that provides a good fit to daily precipitation depths has long been a topic interest".**

*Response:*

We will explain why "Establishing a probability distribution that provides a good fit to daily precipitation depths has long been a topic interest".

**Comment 2: The research objectives are included in the subsection "Precipitation trends and changes", which isn't really logical. Consider restructuring the Introduction, for example, by adding a "Research objectives" subsection.**

*Response:*

Thank you. We will reorganize our introduction so that we state the research objectives explicitly at the end of Introduction section.

**Comment 3: The Introduction is almost half of the paper. Considering shortening it or moving the less essential material to a background subsection.**

*Response:*

Thank you. We will shorten the introduction section in the revised manuscript to focus attention on the distribution of wet-day precipitation.

**Comment 4: Line 267: Regarding "less than "0.01" recordable precipitation," what are the units of the 0.01? Isn't this threshold too low given the detection limit of gauges (approximately 0.25 mm)?**

 *Response:*

 It is common practice in the U.S. to report daily precipitation amounts in inches and

 0.01 inches is commonly considered the detection limit which is approximately

 equivalent to 0.25 mm. We will be sure to emphasize that units are in inches, or

 alternatively, we will transform all units to the metric system.

 **Comment 5: Can you show some maps of the results to reveal what the spatial**

 **patterns in the results look like? Are there any striking differences between,**

 **for example, the temperate southeastern and arid southwestern US?**

 *Response:*

 We will add maps to show the spatial pattern of the results.

 **Comment 6: A Discussion section is missing from the paper.**

 *Response:*

 We will add a 'Discussion' section before the 'Conclusions'.

 **Comment 7: What is the broader significance of the results? Are the results**

 **representative of the rest of the world?**

 *Response:*

 We will add a discussion of the broader significance of the results to the manuscript. In

 short, with increased attention on the impact of climate change, an understanding of the

 distribution of daily wet-day precipitation is paramount for modeling such impacts.

 Importantly, the U.S. continent has extremely broad variations in climatic conditions so

 that the results of our study, which employ very large continental datasets, should be

 illustrative of most other regions of the globe, with the exception of extremely tropical

 and extremely frigid climates, of which there are none in the U.S.

---

## Author Comment (AC3) · 21 Jun 2018

**"The Probability Distribution of Daily Precipitation at the Point and Catchment Scales in the United States" by Lei Ye et al.**

**Response to Referee #3**

We greatly appreciate you for your constructive comments and suggestions. Our responses to the comments are listed below.

**Comment 1: The 'Introduction' section describes in great detail the vast literature related to the topics of (1) stochastic precipitation modeling, (2) precipitation frequency analysis, and (3) precipitation tends and climate changes. In this thorough review it is apparent that the Pearson Type III (P3) distribution has not been considered as a candidate distribution to describe wet-day, AMS or PDS daily precipitation series. Yet the consideration of the P3 distribution is largely explored in this paper. Recommend the authors add why they believe the P3 is an appropriate distribution for the extreme values of rainfall.**

*Response:*

The two parameter Gamma distribution is the most widely used distribution of daily rainfall in previous studies. Therefore it is only natural that one should also consider fitting a three parameter version of the Gamma distribution, known as the P3 distribution to daily rainfall amounts. Given that hundreds of studies have assumed the Gamma distribution, we were very surprised to find so little attention given to the three parameter version of the Gamma distribution, known as the P3 distribution. This is one of the most important contributions of our paper, bringing this fact to light.

**Comment 2: Similar to Referee #2, I believe too much detail is presented in the 'Introduction' section. The lengthy discussion doesn't add to the flow of the paper. Recommend reducing the literature review discussion, highlighting the important studies related to the topics in this paper and refer the reader to**

**Table 1 for a more thorough review of previous studies.**

*Response:*

As you said, the Introduction does indeed account for a relatively large proportion of our paper. We will shorten it to the right proportion and adopt your suggestion (reducing thing literature review discussion, highlighting the important studies related to the topics in this paper and refering the reader to Table 1 for a more thorough review of previous studies.).

**Comment 3: Similar to Referee #2, a 'Discussion' section is missing in this paper**

**and I recommend it be added.**

*Response:*

We will add a 'Discussion' section before the 'Conclusions'.

**Comment 4: 234-239 is interpretive and describes the findings of this paper. This**

**should be moved to the 'Discussion' and/or 'Conclusions' sections. Similarly,**

**the last sentence in the 'Introduction' section (lines 243-245) is interpretive**

**and should be moved to the 'Discussion' and/or 'Conclusions' sections.**

*Response:*

We will move the contents of lines 234-239 and lines 243-245 to the 'Discussion'.

---

## Author Response (AR1)

Response to Reviewers

**Title:** The Probability Distribution of Daily Precipitation at the Point and Catchment

Scales in the United States

**Manuscript ID:** hess-2015-92

**Authors:** Lei Ye, Lars S. Hanson, Pengqi Ding, Dingbao Wang, Richard M.Vogel

Dear Editor:

We greatly appreciate the constructive comments and suggestions provided by the reviewers. We fully agree with the comments on the analysis of complete series of daily precipitation. Thank you for pointing this out, and we have deleted all the results pertaining to the complete rainfall series. Our detailed responses to the comments are listed below.

**Reviewer #1**

**Comment 1: It only makes sense to advance a single continuous pdf for the wet day case, regardless of where the data arises from, if one adds zeros, the Lmoments will always land exactly on the Pearson Type III curve**

*Response:*

Thank you so much for pointing this out. In this revised manuscript, we have eliminated all the results relating to the 'all day' or X>=0 conditions, focusing only on the probability distribution of wet-day precipitation.

**Comment 2: Lines 363-365: "demonstrating that the parameter Gamma distribution cannot describe the tail behavior of full-record series of precipitation as has often been assumed in the past." These lines are just the opportunity for commenting on tail issues. Summary shape statistics are of course affected by the tail behavior but they are not sufficient to reveal in a**

**robust way the behavior of the tail if the whole sample is used (I mean all nonzero values) and not values that belong to the tail. For example in the paper the authors cite (Papalexiou and Koutsoyiannis, 2016) after the fitting using L-moments various measures were proposed in order to compare the fitting in the most extreme value, the largest extremes the whole sample etc. The author can see that the performance of distributions changed, still the GG performed better but the BrXII increased its performance too. I just want to say that indeed this approach can favour specific distributions and exclude others like the G2 the authors mention, yet this is based judging the whole distributional shape properties and it is not really robust to judge on the tail when using the whole nonzero sample. Also other global studies indicated the sub exponential nature of tails focusing on using only "tail" data (Papalexiou et al., 2013; Serinaldi and Kilsby, 2014); the latter was also applied in a seasonal basis, which by the way might be also a nice idea, i.e., the authors to explore seasonal variation.**

*Response:*

Thank you. The sentence has been removed accordingly in the revised manuscript since our analysis of the full-record series of precipitation has been eliminated. The reviewer has suggested a good idea to explore the seasonal variation of the distribution of daily rainfall. The ideas introduced by Papalexiou and Koutsoyiannis (2016) and others concerning the impact of seasons on rainfall distributions could be a future effort. We are aware that the choice of a suitable distribution for modeling rainfall would be quite different if we were to focus our attention on the extreme tail behavior, as is the case for example, when one fits a distribution to the series of annual maxima or peaks above some threshold. However given our interest in wet-day precipitation, most situations of practical relevance and concern are not with extreme rainfall, thus we focus on the complete series of wet-day amounts in this paper, without special attention given to the largest values.

**Comment 3: The P3 distribution is just the two-parameter Gamma distribution**

**(G2) with an additional location parameter which does not affect the shape**

**characteristics and thus $\tau\tau3$ and $\tau\tau4$. So the curve of P3 shown in $\tau4-\tau\tau3$ ratio**

**plots is the same as the G2. And obviously they have the same tail. The same**

**holds for GPA and GP2 and for any other distribution that has one shape**

**parameter and additional location parameters are added. Maybe to ease the**

**reader, as different formulations can be found in the literature, it would be no**

**harm to add a table of the distributions functions used.**

_Response:_

Thank you. We have added distribution functions into Tables 2 and 3.

**Table 2:** Theoretical probability distributions presented on the L-Kurtosis vs L-Skew
L-moment diagram. _Italicized distributions are special cases of other_
_distributions._

| Distribution | Abbreviation | PDF | Parameters |
|---|---|---|---|
| Generalized Extreme Value Type III | GEV | $f(x)=\dfrac{1}{\eta}\left[1-\left(\dfrac{x-\omega}{\eta}\right)\beta\right]^{\frac{1}{\beta}-1}\exp\left\{-\left[1-\left(\dfrac{x-\omega}{\eta}\right)\beta\right]^{\frac{1}{\beta}}\right\}$ | 3 |
| Generalized Logistic | GLO | $f(x)=\dfrac{be^{-\frac{x-\mu}{\sigma}}}{\sigma\left(1+e^{-\frac{x-\mu}{\sigma}}\right)^{b+1}}$ | 3 |
| Generalized Pareto | GPA | $f(x)=\dfrac{1}{\sigma}\left(1+\dfrac{\xi(x-\mu)}{\sigma}\right)^{(-1/\xi-1)}$ | 3 |
| Lognormal | LN3 | $f(x)=\dfrac{1}{\sqrt{2\pi}(x-\alpha)\sigma}\exp\left[-\dfrac{1}{2\sigma^2}\left(\ln(x-\alpha)-\mu^2\right)\right]$ | 3 |
| Pearson Type III | P3 | $f(x)=\dfrac{p/a^d}{\Gamma(d/p)}x^{d-1}e^{-(x/a)^p}$ | 3 |
| _Exponential_ | E | $f(x)=\begin{cases}\lambda e^{-\lambda x},x\geq0\\ 0,x<0\end{cases}$ | 2 |
| _Gumbel_ | G | $f(x)=\dfrac{1}{\beta}e^{-\left(z+e^{-z}\right)},z=\dfrac{x-\mu}{\beta}$ | 2 |
| _Normal_ | N | $f(x)=\dfrac{1}{\sqrt{2\pi\sigma^2}}e^{-\frac{(x-\mu)^2}{2\sigma^2}}$ | 2 |

| | | | |
|---|---|---|---|
| *Logistic* | L | $f(x) = \dfrac{e^{-\frac{x-\mu}{s}}}{s\left(1+e^{-\frac{x-\mu}{s}}\right)^2}$ | 2 |
| *Uniform* | U | $f(x) = \begin{cases} \dfrac{1}{b-a}, & a < x < b \\ 0, & x < a \text{ or } x > b \end{cases}$ | 1 |

**Table 3:** Theoretical probability distributions presented on the L-Cv vs L-Skew L-moment diagram.

| Distribution | Abbreviation | PDF | Parameters |
|---|---|---|---|
| Gamma | G2 | $f(x) = \dfrac{\beta^\alpha x^{\alpha-1} e^{-\beta x}}{\Gamma(\alpha)}$ | 2 |
| Generalized Pareto | GP2 | $f(x) = \dfrac{1}{\sigma}\left(1+\dfrac{\xi x}{\sigma}\right)^{(-1/\xi-1)}$ | 2 |
| Lognormal | LN2 | $f(x) = \dfrac{1}{x\sigma\sqrt{2\pi}} e^{-\frac{(\ln x-\mu)^2}{2\sigma^2}}$ | 2 |
| Weibull | W2 | $f(x) = \dfrac{k}{\phi}\left(\dfrac{x}{\phi}\right)^{k-1} \exp\left\{-\left[\frac{k}{\phi}\right]^k\right\}$ | 2 |

**Comment 4: The Weibull distribution could also be added in the analysis as a representative of distributions with stretched exponential tails.**

*Response:*

Thanks. We have added the Weibull (W2) distribution in the figure below and found that W2 was not competitive with the three potential candidate pdfs G2, P2, and KAP. Again, we emphasize that our interest is in the entire distribution, without focusing attention on the extreme tail behavior.

[Figure]

**Comment 5: When we use distributions with a location parameter to describe a positive variable like the nonzero precipitation this parameter might end far from zero or even negative indicating a lower bound. So, this distribution cannot be used in stochastic modelling of precipitation as it will result in inconsistent values. It would be interesting the authors to actually show some box plots of the estimated parameters.**

*Response:*

Thank you so much for your comment. We agree with you that some parameters can be very interesting. However, the focus of the paper is on the choice of distribution for daily precipitation. In order to save the length of the paper, we don't discuss the parameters here. We will study the parameters in the future work.

**Comment 6: The principle of parsimony should always be applied. If the authors, generate samples from a 4-parameter distribution like the kappa and try to estimate a posteriori the parameters, even for the sample sizes used here, they will find a huge variability that makes, in my opinion, the operational use of 4-parameter distributions quite risky. Of course a 4-parameter distribution like the kappa has a great flexibility, yet this does imply that a better fitting**

**to an observed sample is a better choice to extrapolate values for large return**

**periods.**

*Response:*

We fully agree that in most applications in hydrology, the principle of parsimony is absolutely paramount, due to the short samples available for fitting distributions.

However, in this application, with samples sizes in the tens of thousands, concerns over parsimony are not nearly as critical, even when estimating the KAP distribution. This fact has been shown nicely in the recent paper by Blum et al (2017) where they demonstrated, for similarly length samples of daily streamflow the sampling properties of estimated Lmoments from synthetic samples in their Figure 2.

We do not think we should be advocating a four parameter distribution (KAP) unless absolutely necessary, because it is a much more complex model than may be needed.

We conclude that for representing wet-day precipitation, the Gamma and Pearson-III

distributions are comparable with the 4-parameter Kappa distribution for the areal average precipitation, with P3 only slightly better than G2; however, when the point precipitation is of concern, the Kappa distribution could be the distribution of choice.

**Comment 7: The authors, since this is the first large scale study on catchment**

**precipitation, could provide some analysis on the relation of catchment size**

**and distributional shape. As the spatial averaging process will tend to make**

**the distributions more bell-shaped and with thinner tails. This is the**

**explanation of the performance decrease of the heavy-tailed distribution**

**shown in Fig. 7 compared to Fig. 6 (commenting on the Wet-day; full-day**

**results should be modified).**

*Response:*

Thank you for your suggestion. In the revised manuscript, we have explored the relation between catchment size and distribution shape in the Discussion section.

Lines 393-403: "Figure 9 displays the PPCC values of P3 and G2 pdfs versus catchment drainage area for areal average wet-day series. The PPCC values are chosen from 0.99-

1, approximately 96% of catchments are displayed on the figure; the remaining points lie outside the plot domains. It can be seen that for most of the catchments, the PPCC

values for G2 and P3 pdfs are very close, with points corresponding to G2 and P3 pdfs almost overlapping. This is especially true for PPCC values higher than 0.998.   The phenomena clearly indicates that when G2 can well represent the behavior of catchment-scale wet-day precipitation series, P3 also provides very good performance.

However, for the areas where PPCC values are lower than 0.996, the P3 distribution outperforms the G2 distribution for most cases, with a very slight improvement."

[Figure]

**Figure 9:** the PPCC values of P3 and G2 pdfs versus catchment drainage area for areal average wet-
day series.

**Comment 8: Some regions of the USA, mainly in Midwest, show quite intense**

**changes (or maybe natural variability) on extremes. The authors could also**

**comment on that or do a quick extra analysis on the daily precipitation.**

*Response:*

Thank you very much for you suggestion. We have displayed the best distribution functions for areal average wet-day series of the catchments by showing the location on a map.

Lines 405-412: "KAP distribution is the best pdf for large proportion of the catchments especially in the middle of US. P3 distribution occupies the second large proportion of the catchments especially in east-central US. Only a very few catchments can be best represented by G2 distribution. Seen from Figure 10, it seems that the performances of the three pdfs vary greatly. However, as we have seen from previous figures, the differences between the three pdfs for catchments are very small."

[Figure]

**Figure 10:** The spatial map of catchments with the corresponding best distribution functions for
areal average wet-day series.

**Comment 9: I believe the literature should be updated with many other works, e.g.,**

**there are several papers that are using other distributions for daily**

**precipitation, e.g., one that came to mind is the by Wilson and Toumi (2005).**

*Response:*

Thank you. We have updated the literature review with a focus on the distribution of wet-day precipitation amounts. The following papers have been added in the

References.

Chen, J., Brissette, F. P., and Leconte, R.. Downscaling of weather generator
parameters to quantify hydrological impacts of climate change. Climate Research, 51,
185-200, 2012.

Kigobe, M., McIntyre, N., Wheater, H., and Chandler, R.. Multi-site stochastic
modelling of daily rainfall in Uganda. Hydrological sciences journal, 56, 17-33, 2011.

Li, Z., Brissette, F., Chen, J.. Finding the most appropriate precipitation probability
distribution for stochastic weather generation and hydrological modelling in Nordic
watersheds. Hydrological Processes, 27: 3718-3729, 2013.

Wilson, P. S., and Toumi, R.. A fundamental probability distribution for heavy rainfall. Geophysical Research Letters, 32, L14812, 2005.

**Reviewer #2**

**Comment 1: It should probably be explained in the Introduction why "Establishing a probability distribution that provides a good fit to daily precipitation depths has long been a topic interest".**

*Response:*

Thank you for your comment. We have explained why "Establishing a probability distribution that provides a good fit to daily precipitation depths has long been a topic interest" in the introduction section as follows.

Lines 33-36: "Precipitation is paramount in the fields of hydrology, meteorology, climatology, and others. However, long series of precipitation data are not always available; therefore, establishing a probability distribution that provides a good fit to daily precipitation depths has long been a topic interest."

**Comment 2: The research objectives are included in the subsection "Precipitation trends and changes", which isn't really logical. Consider restructuring the Introduction, for example, by adding a "Research objectives" subsection.**

*Response:*

Thank you. We have reorganized the introduction section by adding a "Research objectives" subsection so that we state the research objectives explicitly at the end of the introduction section.

**Comment 3: The Introduction is almost half of the paper. Considering shortening it or moving the less essential material to a background subsection.**

*Response:*

Thank you. We have shortened the introduction section in the revised manuscript to focus on the distribution of wet-day precipitation.

**Comment 4: Line 267: Regarding "less than "0.01" recordable precipitation,"**

**what are the units of the 0.01? Isn't this threshold too low given the detection**

**limit of gauges (approximately 0.25 mm)?**

*Response:*

It is common practice in the US to report daily precipitation amounts in inches and 0.01

inches is commonly considered the detection limit which is approximately equivalent to 0.25 mm. We have explained it in the paper as follows.

Lines 198-200: "The wet-day series were constructed by excluding zero and "trace"

values (those with less than 0.01 inches (approximately equivalent to 0.25 mm)

recordable precipitation)."

**Comment 5: Can you show some maps of the results to reveal what the spatial**

**patterns in the results look like? Are there any striking differences between,**

**for example, the temperate southeastern and arid southwestern US?**

*Response:*

Thank you very much for you suggestion. We have displayed the best distribution functions for areal average wet-day series of the catchments by showing the location on a map.

Lines 405-412: "KAP distribution is the best pdf for large proportion of the catchments especially in the middle of US. P3 distribution occupies the second large proportion of the catchments especially in east-central US. Only a very few catchments can be best represented by G2 distribution. Seen from Figure 10, it seems that the performances of the three pdfs vary greatly. However, as we have seen from previous figures, the differences between the three pdfs for catchments are very small."

[Figure]

**Figure 10:** The spatial map of catchments with the corresponding best distribution functions for areal average wet-day series.

**Comment 6:    A Discussion section is missing from the paper.**

*Response:*

We have added a Discussion section before the Conclusions.

**Comment 7: What is the broader significance of the results? Are the results representative of the rest of the world?**

*Response:*

With increased attention on the impact of climate change, an understanding of the distribution of daily wet-day precipitation is paramount for modeling such impacts. Importantly, the U.S. continent has extremely broad variations in climatic conditions so that the results of our study, which employ very large continental datasets, should be illustrative of most other regions of the globe, with the exception of extremely tropical and extremely frigid climates, of which there are none in the U.S.

**Reviewer #3**

**Comment 1: The 'Introduction' section describes in great detail the vast literature related to the topics of (1) stochastic precipitation modeling, (2) precipitation**

**frequency analysis, and (3) precipitation tends and climate changes. In this thorough review it is apparent that the Pearson Type III (P3) distribution has not been considered as a candidate distribution to describe wet-day, AMS or PDS daily precipitation series. Yet the consideration of the P3 distribution is largely explored in this paper. Recommend the authors add why they believe the P3 is an appropriate distribution for the extreme values of rainfall.**

*Response:*

The two parameter Gamma distribution is the most widely used distribution of daily rainfall in previous studies. Therefore it is only natural that one should also consider fitting a three parameter version of the Gamma distribution, known as the P3 distribution to daily rainfall amounts. Given that hundreds of studies have assumed the Gamma distribution, we were very surprised to find so little attention given to the three parameter version of the Gamma distribution, known as the P3 distribution. This is one of the most important contributions of our paper, bringing this fact to light.

**Comment 2: Similar to Referee #2, I believe too much detail is presented in the 'Introduction' section. The lengthy discussion doesn't add to the flow of the paper. Recommend reducing the literature review discussion, highlighting the important studies related to the topics in this paper and refer the reader to Table 1 for a more thorough review of previous studies.**

*Response:*

As you said, the introduction section indeed accounts for a relatively large proportion of the manuscript. We have shortened the introduction section significantly and refer the reader to Table 1 for a more thorough review of previous studies.

**Comment 3: Similar to Referee #2, a 'Discussion' section is missing in this paper and I recommend it be added.**

*Response:*

We have added a Discussion section before the Conclusions.

**Comment 4: 234-239 is interpretive and describes the findings of this paper. This should be moved to the 'Discussion' and/or 'Conclusions' sections. Similarly, the last sentence in the 'Introduction' section (lines 243-245) is interpretive and should be moved to the 'Discussion' and/or 'Conclusions' sections.**

*Response:*

We have moved the contents of lines 234-239 and lines 243-245 to the Conclusions section.

[revised manuscript text omitted]

---

## Author Response (AR2)

**Response to Referee #1**

**Title:** The Probability Distribution of Daily Precipitation at the Point and Catchment Scales in the United States

**Manuscript ID:** hess-2018-85

**Authors:** Lei Ye, Lars S. Hanson, Pengqi Ding, Dingbao Wang, Richard M.Vogel

Dear Editor and Referees:

Thank you for your comments and suggestions for our paper. We have revised the manuscript based on the comments and our detailed responses are listed below.

**Comment 1: Abstract: Personally, I would avoid a direct comparison of two- (G2) and three-parameter distributions (P3). Especially, since P3 is a generalization of G2 with an additional location parameter then "by definition" always will perform better. So maybe the authors wish to reform these statements.**

*Response:*

We have reformed the statement by deleting the direct comparison of G2 and P3 in Abstract, i.e., the sentence has been revised as:

"Importantly, both Pearson Type-III (P3) and Kappa (KAP) distributions perform very well particularly for point rainfall."

**Comment 2: Heading: In some heading authors are using ":" in the end that is not necessary.**

*Response:*

Corrected.

**Comment 3: Section 1.1: The GG was also used in stochastic modelling of precipitation, see Fig.5 for hourly and Fig.6 for daily in Papalexiou (2018). Actually any distribution that describes well wet-day precipitation (or at any other scale) can be used as this stochastic modelling scheme makes feasible to**

**use any probability distribution and any correlation structure. Additionally, the GG was also used to generate gridded daily precipitation that is consistent with monthly observations (see Fig. 11 in Papalexiou et al., 2018).**

*Response:*

Thanks for your explanation about GG distribution. We have explained the GG distribution and the paper Papalexiou et al. (2018) has been cited in our paper.

**Comment 4: L147: Here is mentioned that there is a consensus in using GEV, G, or GPA, for annual maxima. I would agree for G and GEV but the GPA is mainly used for peak above threshold values. Most of annual maxima samples have a bell-shaped density while the GPA is always J-shaped and of course its origin is from the Pickands–Balkema–de Haan theorem which refers to POT values.**

*Response:*

Thank you. The sentence has been revised as:

"There seems to be a consensus that annual maxima appear to be well approximated by either a GEV or Gumbel probability density function (pdf); while peaks above threshold values are well approximated by a GPA distribution, and the series of wet-day precipitation is well approximated by a G2, GG, W2 or in some cases a mixed exponential distribution."

**Comment 5: L160. Please delete the "generalized" before the "Burr type XII". Also just to explain the point made by the authors in the following lines regarding L-skew/L-kurt diagrams for the GG and BrXII(of course they do not have to add this in the manuscript). The truth is that they exist and I had formed the theoretical spaces similar to L-CV/L-skew but the problem was that the space was not "one-to-one", I mean that for different shape-parameters points the same point in the L-skew/L-kurt emerged. So, it was not possible to depict the theoretical space without avoiding overlapping.**

*Response:*

We have deleted the "generalized" before the "Burr type XII" and thanks for your kind explanation regarding GG and BrXII.

**Comment 6: L217: Since the authors are not including zeros in the analysis (and that is correct) there is no point to mention zeros a possible explanation of the skewness. So, the distribution of daily totals (of wet days of course) is highly skewed due to small nonzero values and high variance, i.e., since its bounded at zero and there is a high frequency of close-to-zero values and the variance is high then inevitable this creates the positive skewness.**

*Response:*

We have rewritten the sentence as:

"The distribution of wet-day series of precipitation is highly skewed due to the large proportion of small non-zero values and high variance."

**Comment 7: L265: use italics for p, n and i, to display the equation properly.**

*Response:*

Corrected.

**Comment 8: Section 4.1.2: I think the explanation why points shift towards the G2 or P3 curve is due to the areal averaging process which lowers the variance and the kurtosis. I am a bit surprised though that I do not see lower values of skewness.**

*Response:*

We agree with your opinion that areal averaging process lowers the variance and the kurtosis.

**Comment 9: L321: At least in the title (4.2) I would suggest not using the abbreviation PPCC**

*Response:*

The abbreviation "PPCC" in title has been spelled out.

**Comment 10: L388: I am sure that the statement "the KAP is required to capture the tail behavior…" is a bit risky. I stress again that the Kappa distribution is a four parameter distribution and therefore has a huge flexibility and can fit to anything. Yet this does not imply that it can capture the behavior of the tail; the extrapolations could be very risky especially if estimation is based on small samples.**

*Response:*

The sentence has been revised as:

"From the L-moment diagrams and PPCC comparisons we concluded that KAP can better capture the tail behavior of point wet-day series, though both P3 and G2 can provide reasonable approximations in many situations."

**Comment 11: Table 2: An optional suggestion: there is a huge mess in the literature regarding the parameter symbols used in distribution which creates confusion. For example, here η, ω, ξ, μ, σ ,p, d, λ, β, b, φ, κ etc are used. There are only 3 type of parameters, location, scale and shape. I personally follow a clear and simple convention, i.e., α for location, β for scale, and γ for shape; if more than one parameter of the same type exist I use indices (γ_1, γ_2). This allows the reader to know the function of the parameters. Of course there are classical cases like the normal distribution where is standard to μ and σ due to the link with mean an sd. Please check the equations: the P3 authors show is something between Weibull and Gamma. To simplify reading authors can use exp instead of e. Check how equation are depicted, probably authors introduced eqs as pictures that creates distortion and unequal font size. Authors are not showing the Kappa distribution as it forms a space, but for completeness I would suggest to provide its formula.**

*Response:*

Thanks. The equations have been revised.

**Comment 12: Table 3: The Generalized Pareto is not the GP2, either correct the name or the equation. Please, correct the formula of the Weibull distribution, you are using exp and then superscript; no need for {} or [].**

*Response:*

The Generalized Pareto in the paper is the two-parameter Generalized Pareto distribution. We have revised the equations of GP2 and Weibull.

**Comment 13: Figure 2: X-axis maybe days? The same holds for Fig.3**

*Response:*

The labels for X-axis in Figure 2 and Figure 3 have been changed to "Number of days".

**Comment 14: Fig 5. I mentioned that before. The theoretical lines of G2 = P3, GP2 = GPA, LN2 = LN3 for Lskew-Lkurt diagrams. Authors can mention that if they wish.**

*Response:*

We have added an explanation in the revised paper.

"It should be noted that the P3 distribution is the two-parameter G2 with an additional location parameter which does not affect the shape characteristics and thus the theoretical curve of P3 shown in Figure 5 is the same as the G2. The same holds for GPA and GP2 and for LN2 and LN3."